



# Production of HONO from heterogeneous uptake of NO₂ on illuminated TiO₂ aerosols measured by Photo-Fragmentation Laser Induced Fluorescence

Joanna E. Dyson[1], Graham A. Boustead[1], Lauren T. Fleming[1], Mark Blitz[1,2], Daniel Stone[1],
Stephen R. Arnold[3], Lisa K. Whalley[1,2], Dwayne E. Heard[1]*

[1] *School of Chemistry, University of Leeds, LS2 9JT, UK.*

[2] *National Centre of Atmospheric Science, University of Leeds, LS2 9JT, UK.*

[3] *School of Earth and Environment, University of Leeds, LS2 9JT, UK.*

*Corresponding Author. Email: D.E.Heard@leeds.ac.uk*

**Abstract**

The rate of production of HONO from illuminated TiO₂ aerosols in the presence of NO₂ was measured using an aerosol flow tube coupled to a photo-fragmentation laser induced fluorescence detection apparatus. The reactive uptake coefficient of NO₂ to form HONO, $\gamma_{NO_2 \rightarrow HONO}$, was determined for NO₂ mixing ratios in the range 34 – 400 ppb, with $\gamma_{NO_2 \rightarrow HONO}$ spanning the range $(9.97 \pm 3.52) \times 10^{-6}$ to $(1.26 \pm 0.17) \times 10^{-4}$ at a relative humidity of $15 \pm 1$ % and for a lamp photon flux of $(1.63 \pm 0.09) \times 10^{16}$ photons cm⁻² s⁻¹ (integrated between 290 and 400 nm), which is similar to values of ambient actinic flux at midday. $\gamma_{NO_2 \rightarrow HONO}$ increased as a function of NO₂ mixing ratio at low NO₂ before peaking at $(1.26 \pm 0.17) \times 10^{-4}$ at 51 ppb NO₂ and then sharply decreasing at higher NO₂ mixing ratios, rather than levelling off which would be indicative of surface saturation. The dependence of HONO production on relative humidity was also investigated, with a peak in production of HONO from TiO₂ aerosol surfaces found at ~25 % RH. Possible mechanisms consistent with the observed trends in both the HONO production and reactive uptake coefficient were investigated using a zero-dimensional kinetic box model. The modelling studies supported a mechanism for HONO production on the aerosol surface involving two molecules of NO₂, as well as a surface HONO loss mechanism which is dependent upon NO₂. In a separate experiment, significant production of HONO was observed from illumination of mixed nitrate/TiO₂ aerosols in the absence of NO₂. However, no statistically significant production of HONO was seen from the illumination




of pure nitrate aerosols. The rate of production of HONO observed from mixed nitrate/TiO$_2$
aerosols was scaled to ambient conditions found at the Cape Verde Atmospheric Observatory
(CVAO) in the remote tropical marine boundary layer. The rate of HONO production from
aerosol particulate nitrate photolysis containing a photocatalyst was found to be similar to the
missing HONO production rate necessary to reproduce observed concentrations of HONO at
CVAO. These results provide evidence that particulate nitrate photolysis may have a significant
impact on the production of HONO and hence NO$_x$ in the marine boundary layer where mixed
aerosols containing nitrate and a photocatalytic species such as TiO$_2$, as found in dust, are
present.
**1    Introduction**
A dominant source of OH radicals in polluted environments is the photolysis of nitrous acid
(HONO) (Platt et al., 1980;Winer and Biermann, 1994;Harrison et al., 1996;Alicke et al.,
2002;Whalley et al., 2018;Crilley et al., 2019;Lu et al., 2019;Slater et al., 2020;Whalley et al.,
2020). During a recent study in Winter in central Beijing, HONO photolysis accounted for over
90 % of the primary production of OH averaged over the day (Slater et al., 2020). Oxidation
by OH radicals is the dominant removal mechanism for many tropospheric trace gases, such as
tropospheric methane, as well as the formation of secondary species, including tropospheric
ozone (Levy, 1971), nitric and sulphuric acids which condense to form aerosols, and secondary
organic aerosols. Understanding the formation of HONO in highly polluted environments is
crucial to fully understand both the concentration and distribution of key atmospheric radical
species, as well as secondary products in the gas and aerosol phases associated with climate
change and poor air quality.
Atmospheric concentrations of HONO range from a few pptv in remote clean environments
(Reed et al., 2017) to more than 10 ppb in highly polluted areas such as Beijing (Crilley et al.,
2019). The main gas-phase source of HONO in the troposphere is the reaction of nitric oxide
(NO) with the OH radical. HONO has also been shown to be directly emitted from vehicles
(Kurtenbach et al., 2001;Li et al., 2008), for which the rate of emission is often estimated as a
fraction of known NO$_x$ (NO$_2$+NO) emissions. Many heterogeneous HONO sources have also
been postulated including the conversion of nitric acid (HNO$_3$) on ground or canopy surfaces
(Zhou et al., 2003;George et al., 2005), bacterial production of nitrite on soil surfaces (Su et
al., 2011;Oswald et al., 2013) and, more recently, particulate nitrate photolysis, thought to be





an important source in marine environments (Ye et al., 2016;Reed et al., 2017;Ye et al.,
2017a;Ye et al., 2017b). Rapid cycling of gas-phase nitric acid to gas-phase nitrous acid via
particulate nitrate photolysis in the clean marine boundary layer has been observed during the
2013 NOMADSS aircraft measurements campaign over the North Atlantic Ocean (Ye et al.,
2016). Ground-based measurements of HONO made at Cape Verde in the tropical Atlantic
Ocean (Reed et al., 2017) provided evidence that a mechanism for renoxification in low $NO_x$
areas is required (Reed et al., 2017;Ye et al., 2017a).
Recent model calculations show a missing daytime source of HONO, which is not consistent
with known gas-phase production mechanisms, direct emissions or dark heterogeneous
formation (e.g. prevalent at night). It has been suggested that this source could be light driven
and dependent on $NO_2$ (Kleffmann, 2007;Michoud et al., 2014;Spataro and Ianniello, 2014;Lee
et al., 2016).
It is estimated that between 1604 and 1960 Tg $yr^{-1}$ of dust particles are emitted into the
atmosphere (Ginoux et al., 2001). Titanium dioxide ($TiO_2$) is a photocatalytic compound found
in dust particles at mass mixing ratios of between 0.1 and 10 % depending on the location the
particles were suspended (Hanisch and Crowley, 2003). When exposed to UV light ($\lambda < 390$
nm) $TiO_2$ promotes an electron ($e_{CB}^-$) from the conduction band to the valence band leaving
behind a positively charged hole ($h_{VB}^+$) in the valence band (Chen et al., 2012):

$$TiO_2 + h\nu \rightarrow e_{CB}^- + h_{VB}^+ \qquad\qquad (R1)$$

which can then lead to both reduction and oxidation reactions of any surface adsorbed gas-
phase species such as $NO_2$ leading to HONO.
In previous studies of the reaction of $NO_2$ on $TiO_2$ aerosol surfaces, HONO was observed as a
major gas-phase product (Gustafsson et al., 2006;Dupart et al., 2014). Gustafsson *et al.*, (2006)
observed a yield of gas-phase HONO of ~ 75 % (for each $NO_2$ removed), and showed the rate
of the photoreaction of $NO_2$ on pure $TiO_2$ aerosols depended on relative humidity, emphasising
the superhydrophilic nature of $TiO_2$ surfaces under UV irradiation. Dupart *et al.* (2014) also
reported a relative humidity dependence of the uptake of $NO_2$ onto Arizona Test Dust
containing $TiO_2$ with the main gas-phase products measured being NO and HONO, with a
HONO yield of 30 % in experiments with 110 ppb $NO_2$. Dupart et al. (2014) postulated the
following mechanism of HONO production, which is consistent with the formation of the $NO_2^-$
anion seen in a previous study on $TiO_2$ surfaces (Nakamura et al., 2000):

$$h_{VB}^+ + H_2O \rightarrow H^+ + OH \qquad\qquad (R2)$$

$$e_{CB}^- + O_2 \rightarrow O_2^- \qquad\qquad (R3)$$

$$NO_2 + O_2^- \,(or\, e_{CB}^-) \rightarrow NO_2^- + O_2 \qquad\qquad (R4)$$

$$NO_2^- + H^+ \rightarrow HONO \qquad\qquad (R5)$$

$$NO_2 + OH \rightarrow HNO_3 \qquad\qquad (R6)$$

In areas with high mineral dust loading, such as desert regions, far from anthropogenic sources,
$NO_2$ concentrations are typically low. However, when dust is transported to urban areas, this
source of HONO may become significant. One study reported that $TiO_2$ composed 0.75-1.58
µg m$^{-3}$ when aerosol loadings were 250-520 µg m$^{-3}$ over the same time period in southeast
Beijing, when air had been transported from the Gobi desert (Schleicher et al., 2010).
In this study, the production of HONO on the surface of $TiO_2$ particles in the presence of $NO_2$
is investigated as a function of $NO_2$ mixing ratio, aerosol surface area density and relative
humidity using an aerosol flow tube system coupled to a photo-fragmentation laser induced
fluorescence detector (Boustead, 2019). The uptake coefficient of $NO_2$ to generate HONO is
then determined, and a mechanistic interpretation of the experimental observations is
presented. The production of HONO directly in the absence of $NO_2$ from the illumination of a
mixed sample of nitrate and $TiO_2$ aerosol is also presented. Using a similar apparatus, previous
work had showed that $TiO_2$ particles produce OH and $HO_2$ radicals directly under UV
illumination (Moon et al., 2019). The atmospheric implications of these results and the role of
photo-catalysts for the formation of HONO are also discussed.
**2 Method**
**2.1 Overview of the Experimental Setup**
The production of HONO from illuminated aerosol surfaces is studied using an aerosol flow
tube system coupled to a photo-fragmentation laser induced fluorescence (PF-LIF) cell which
allows the highly sensitive detection of the OH radical formed through photo-fragmentation of
HONO into OH and NO followed by Laser-Induced Fluorescence (LIF) detection at low
pressure. The experimental setup used in this investigation is described in detail in (Boustead,
2019), therefore only a brief description of the setup is given here. A schematic of the
experimental setup is shown in Figure 1.

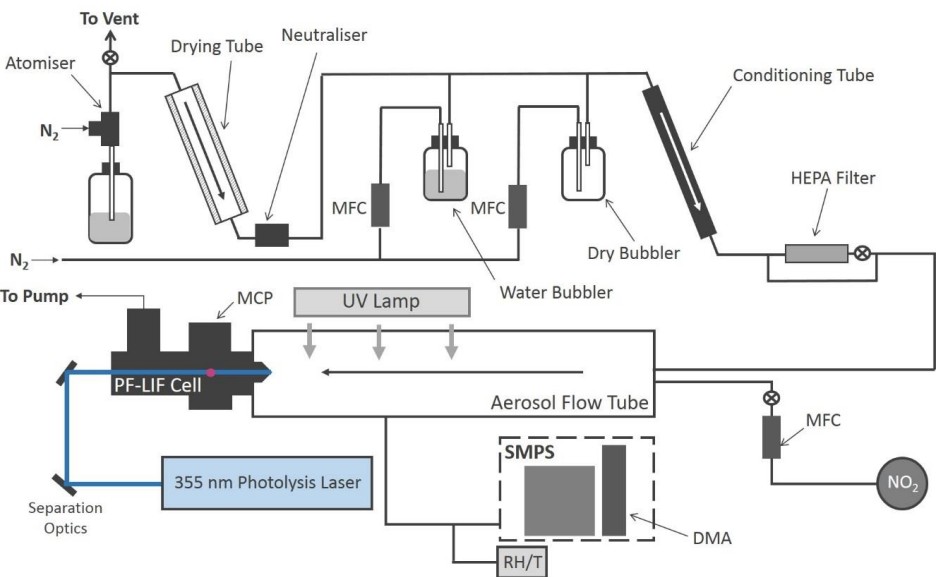

**Figure 1.** Schematic of the Leeds aerosol flow tube system coupled to a laser-fragmentation laser-induced fluorescence detector for HONO. The paths of the 355 nm (blue) and 308 nm (purple, depicted as travelling out of the page perpendicular to the 355 nm light) light are also shown. CPC: condensation particle counter; DMA: differential mobility analyser; HEPA: high efficiency particle air filter; FAGE: fluorescence assay by gas expansion; MCP: microchannel plate photomultiplier; MFC: mass flow controller; RH/T: relative humidity/ temperature probe; SMPS: scanning mobility particle sizer.


All experiments were conducted at room temperature (295 ± 3 K) using nitrogen (BOC, 99.998
%) or air (BOC, 21 ± 0.5 % $O_2$) as the carrier gas. A humidified flow of aerosols, ~ 6 lpm (total
residence time of 104 s in the flow tube), was introduced through an inlet at the rear of the
aerosol flow tube (Quartz, 100 cm long, 5.75 cm ID) which was covered by a black box to
eliminate the presence of room light during experiments. A 15 W UV lamp (XX-15LW Bench
Lamp, $\lambda_{peak}$=365 nm) was situated on the outside of the flow tube to illuminate aerosols and
promote the production of HONO (half the length of the flow tube was illuminated leading to
an illumination time of 52 s). The concentration of HONO is measured by PF-LIF with
sampling from the end of the flow tube via a protruding turret containing a 1 mm diameter
pinhole, through which the gas exiting the flow tube was drawn into the detection cell at 5 lpm.
The detection cell was kept at low pressure, ~ 1.5 Torr, using a rotary pump (Edwards, E1M80)
in combination with a roots blower (Edwards, EH1200). All gas flows in the experiment were
controlled using mass flow controllers (MKS and Brooks). The relative humidity (RH) and
temperature of the aerosol flow was measured using a probe (Rotronics HC2-S, accuracy ±1





% RH) the former calibrated against the H$_2$O vapour concentration measured by a chilled
mirror hygrometer (General Eastern Optica), in the exhaust from the flow tube.

## 2.2  Aerosol generation and detection

Solutions for the generation of TiO$_2$ aerosol solutions were prepared by dissolving 5 g of
titanium dioxide (Aldrich Chemistry 718467, 99.5% Degussa, 80 % anatase: 20 % rutile) into
500 ml of milli-Q water. Polydisperse aerosols were then generated from this solution using an
atomiser (TSI model 3076) creating a 1 lpm flow of TiO$_2$ aerosol particles in nitrogen hereafter
referred to as the aerosol flow. This aerosol flow was then passed through a silica drying tube
(TSI 3062, capable of reducing 60 % RH incoming flow to 20 % RH) to remove water vapour,
then passed through a neutraliser to apply a known charge distribution and reduce loss of
aerosols to the walls. After the neutraliser the aerosol flow was mixed with both a dry and a
humidified N$_2$ flow (controlled by MFCs) to regulate the relative humidity of the system by
changing the ratio of dry to humid nitrogen flows. A conditioning tube was then used to allow
for equilibration of water vapour adsorption and re-evaporation to and from the aerosol surfaces
for the chosen RH, which was controlled within the range ~10-70 % RH. A portion of the
aerosol flow was then passed through a high efficiency particle filter (HEPA) fitted with a
bypass loop and bellows valve allowing control of the aerosol number concentration entering
the aerosol flow tube. Previous studies (George et al., 2013;Boustead, 2019) have shown the
loss of aerosol to the walls of the flow tube to be negligible. Aerosol size distributions were
measured for aerosols exiting the flow tube using a scanning mobility particle sizer (SMPS,
TSI 3081) and a condensation particle counter (CPC, TSI 3775) which was calibrated using
latex beads. Any aerosol surface area not counted due to the upper diameter range of the
combined SMPS/CPC (14.6 – 661.2 nm, sheath flow of 3 lpm, instrumental particle counting
error of 10-20 %) was corrected for during analysis by assuming a lognormal distribution,
which was verified for TiO$_2$ aerosols generated in this manner (Matthews et al., 2014).
However, the majority of aerosols, >90 %, had diameters in the range that could be directly
detected. Previously, we had determined by imaging the aerosols using a scanning electron
microscope (SEM) that the particles were spherical (Moon et al., 2018). In addition to the
experiments with single-component TiO$_2$, mixed ammonium nitrate/TiO$_2$ and single-
component ammonium nitrate aerosols were also generated using the atomiser for
investigations of HONO production from nitrate aerosols without NO$_2$ present. An example of
an aerosol size distribution from this work for single-component ammonium nitrate aerosols,
mixed ammonium nitrate/TiO$_2$ and single-component TiO$_2$ aerosols is shown in Figure 2.



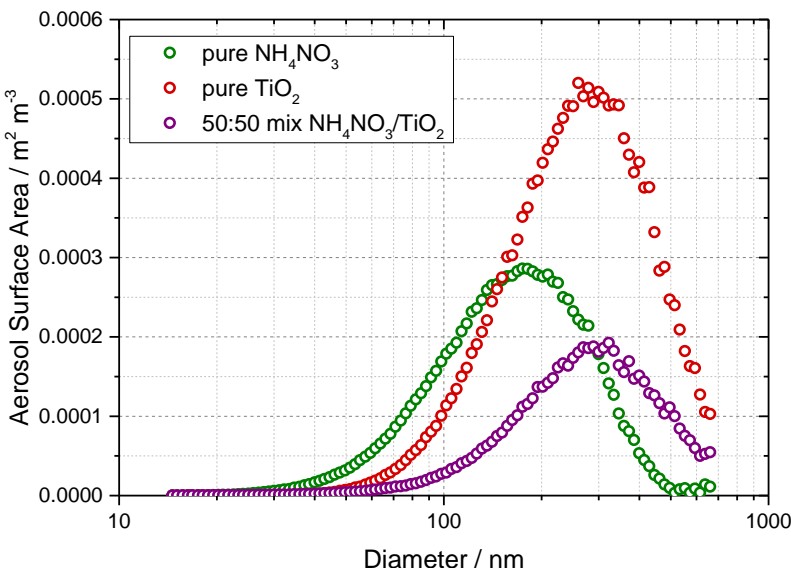

**Figure 2** Typical aerosol surface area distribution for pure ammonium nitrate aerosols (green) and pure $TiO_2$
aerosols (red) and 50:50 mixed nitrate/$TiO_2$ aerosols (purple) measured after the flow tube.

## 2.3 Detection of HONO


As HONO is not directly detectable via LIF, it was necessary to fragment the HONO produced
into OH and NO (Liao et al., 2006), with detection of OH via LIF. A 355 nm photolysis laser
(Spectron Laser Systems, SL803) with a pulse repetition frequency (PRF) of 10 Hz and pulse
duration ~ 10 ns was used to fragment HONO into OH. This fragmentation wavelength was
chosen as HONO has a strong absorption peak at ~ 355 nm leading to the breakage of the HO-
NO bond to form NO and OH in their electronic ground states (Shan et al., 1989). A Nd:YAG
pumped dye probe laser (JDSU Q201-HD, Q-series, Sirah Cobra Stretch) with a PRF of 5000
Hz, was used for the detection of OH via the fluorescence assay by gas expansion (FAGE)
technique which employs the expansion of gas through a small pinhole into the detection cell.
The OH radical was measured using on-resonance detection by LIF via the excitation of the
$A^2\Sigma^+$ (v' = 0) $\leftarrow X^2\Pi_i$ (v'' = 0) Q$_1$(2) transition at 308 nm (Heard, 2006). A multi-channel plate
(MCP) photomultiplier (Photek, MCP 325) equipped with an interference filter at 308 nm (Barr
Associates, 308 nm. FWHM – 8 nm, ~50 % transmission) was used to measure the fluorescence
signal. A reference OH cell in which a large LIF signal could be generated was utilised to



ensure the wavelength of the probe laser remained tuned to the peak of the OH transition at
308 nm. OH measurements are taken both before and after each photolysis laser pulse allowing
measurement of any OH already present in the gas flow to be determined as a background
signal for subtraction. The OH generated from HONO photolysis was measured promptly
(~800 ns) after the 355 nm pulse to maximise sensitivity to OH before it was spatially diluted
away from the measurement region (Boustead, 2019). Offline measurements, with the probe
laser wavelength moved away from the OH transition (by 0.02 nm), were taken to allow the
signal generated from detector dark counts and scattered laser light to be measured and
subtracted from the online signal. To determine an absolute value of the HONO concentration,
[HONO], a calibration was performed, in order to convert from the HONO signal, $S_{HONO}$, using
$S_{HONO} = C_{HONO}$ [HONO], as described fully in (Boustead, 2019). A glass calibration wand was
used to produce OH and $HO_2$ in equal concentrations from the photolysis of water vapour at
185 nm:

$$H_2O + h\nu \xrightarrow{\lambda=185\,nm} OH + H \qquad\qquad (R7)$$

$$H + O_2 + M \rightarrow HO_2 + M \qquad\qquad (R8)$$

An excess flow of NO was then added to generate HONO which was then detected as OH in
the cell. The excess flow of NO (BOC, 99.5 %) ensures rapid and complete conversion of OH
and $HO_2$ to HONO. The concentration of OH and $HO_2$ produced, and therefore the amount of
HONO produced in the wand, is calculated using:

$$[OH] = [HO_2] = [H_2O]\,\sigma_{H2O}\,\phi_{OH}\,F\,t \qquad\qquad (1)$$

where [$H_2O$] is the concentration of water vapour in the humidified gas flow, $\sigma_{H2O}$ is the
absorption cross section of $H_2O$ at 185 nm (7.14 × 10$^{-20}$ cm$^2$ molecule$^{-1}$ (Cantrell et al., 1997),
$\phi_{OH}$ is the quantum yield of OH for the photo-dissociation of $H_2O$ at 185 nm (=1), $F$ is the
lamp flux and $t$ is the irradiation time (the product of which is determined using ozone
actinometry (Boustead, 2019).
A typical value of the calibration factor was $C_{HONO} = (3.63 \pm 0.51) \times 10^{-9}$ counts mW$^{-1}$ for $N_2$,
leading to a calculated limit of detection of 12 ppt for a 50 s averaging period and a signal-to-
noise ratio (SNR) of 1 (Boustead, 2019). The typical error in the HONO concentration was
15% at 1σ, determined by the error in the calibration.



## 2.4  Experimental procedure and data analysis

The experiments were performed with a minimum flow of 6 lpm through the aerosol flow tube giving a Reynolds number of ~ 150 which ensured a laminar flow regime. The HONO signal, converted to an absolute concentration using a calibration factor, was measured over a range of aerosol surface area densities, both in the presence and absence of illumination, and background measurements without aerosols present, were also performed.

The HONO signal originates from several sources: the illuminated aerosol surface; the illuminated quartz flow tube walls; dark reactions on aerosol surfaces; dark reactions on the flow tube surface and finally from impurities in the $NO_2$ (Sigma Aldrich, >99.5 %, freeze pump thawed to further remove any remaining NO or $O_2$) and $N_2$ flows (either HONO itself or a species which photolyses at 355 nm to give OH). Of interest here is the HONO production from both dark and illuminated aerosol surfaces which is atmospherically relevant. Following transit through the flow tube, and in the presence of $NO_2$, the total concentration of HONO measured by the PF-LIF detector is given by:

$$[\text{HONO}] = [\text{HONO}]_{illuminated\ aerosols} + [\text{HONO}]_{illuminated\ walls}$$
$$+ [\text{HONO}]_{dark\ aerosols} + [\text{HONO}]_{dark\ walls} + [\text{HONO}]_{impurities} \tag{2}$$

Any HONO seen without the presence of aerosol was therefore due to HONO impurities in the $N_2$ or $NO_2$ gas, the dark production of HONO from the flow tube walls or from the production of HONO from the illuminated reactor walls, which may include production from $TiO_2$ aerosols coating the flow tube in the presence of $NO_2$. This background HONO concentration depended on the experimental conditions and on how recently the flow tube and PF-LIF cell had been cleaned to remove any build-up of $TiO_2$ deposits. However, the build-up of $TiO_2$ on the flow tube walls was relatively slow and back-to-back measurements were made in the presence and absence of aerosols to obtain an accurate background. Even though the aerosol surface area density (~0.02 $m^2\ m^{-3}$) was small compared to the surface area density of the reactor walls (35 $m^2\ m^{-3}$), very little HONO signal was produced without the presence of aerosols, and was always subtracted from the signal in the presence of aerosols. The HONO signal was measured both with the lamp on and off for each aerosol surface area density to investigate the production of HONO from illuminated aerosol surfaces. The HONO signal was averaged over 50 s (average of 500 of the 355 nm photolysis laser pulses with a PRF of 10 Hz). Once aerosols were introduced into the flow tube system a period of ~ 30 min was allowed for equilibration and the measured aerosol surface area density to stabilise. In general, the relative





humidity of the system was kept constant at RH ~ 15 % for all experiments investigating
HONO production as a function of $NO_2$ mixing ratio over the range 34 - 400 ppb. In a number
of experiments, however, RH was varied in the range ~12-37 %.
The mixing ratio of $NO_2$ entering the flow tube was calculated using the concentration of the
$NO_2$ in the cylinder and the degree of dilution. The $NO_2$ mixing ratio within the cylinder was
determined using a commercial instrument based on UV-Vis absorption spectroscopy (Thermo
Fisher 42TL, limit of detection 50 pptv, precision 25 pptv) For each individual experiment, the
mixing ratio of $NO_2$ was kept constant (within the range 34 – 400 ppb) and the aerosol surface
area density was varied from zero up to a maximum of 0.04 $m^2\,m^{-3}$.  In order to obtain the
HONO produced from illuminated aerosol surfaces in the flow tube for a given mixing ratio of
$NO_2$. As well as subtraction of any background HONO, a correction must be made for any loss
of HONO owing to its photolysis occurring within the flow tube.
In order to determine the rate of photolysis of HONO, the rate of photolysis of $NO_2$ was first
determined using chemical actinometry, and the known spectral output of the lamp and the
literature values of the absorption cross-sections and photo-dissociation quantum yields for
$NO_2$ and HONO were used to determine the rate of photolysis of HONO. When just flowing
$NO_2$ in the flow tube, the loss of $NO_2$ within the illuminated region is determined only by
photolysis and is given by:

$$-\frac{d[NO_2]}{dt} = j(NO_2)[NO_2] \qquad (3)$$

where $j(NO_2)$ is the photolysis frequency of $NO_2$ for the lamp used in these experiments. From
the measured loss of $NO_2$ in the illuminated region, and with knowledge of the residence time,
the photolysis frequency, $j(NO_2)$, was determined to be $(6.43 \pm 0.30) \times 10^{-3}\,s^{-1}$ for the set of
experiments using one lamp to illuminate the flow tube. $j(NO_2)$ is given by:

$$j(NO_2) = \int_{\lambda_1}^{\lambda_2} \sigma_\lambda \phi_\lambda F_\lambda \, d\lambda \qquad (4)$$

where $\lambda_1$ and $\lambda_2$ represent the range of wavelengths over which the lamp emits, and $\sigma_\lambda$ and $\phi_\lambda$
are the wavelength-dependent absorption-cross section and photo-dissociation quantum yield
of $NO_2$, respectively, and $F_\lambda$ is the flux of the lamp at a given wavelength. The flux of the lamp,
the spectral intensity of which was measured using a Spectral Radiometer (Ocean Optics QE-
Pro 500) as a function of wavelength, is shown in Figure 3.



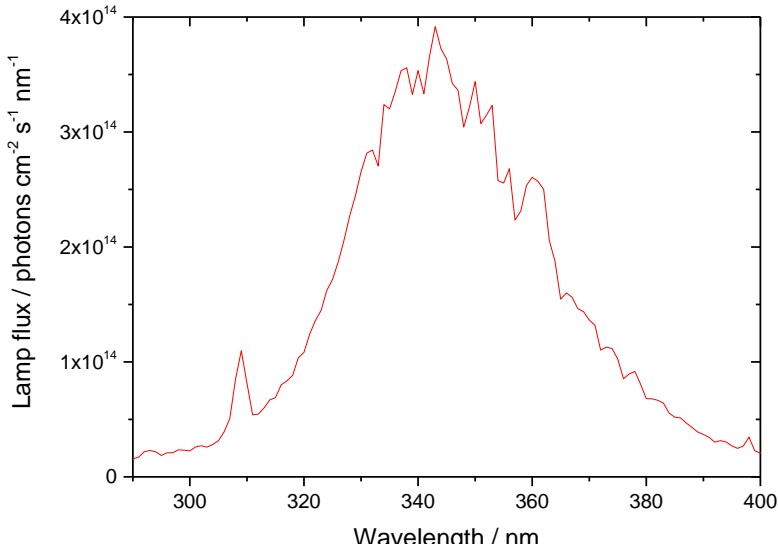

.

**Figure 3.** UVA emission spectrum for the 15 W bench lamp used in these experiments between 290-400 nm. The integrated photon flux over this wavelength range is $(1.63 \pm 0.09) \times 10^{16}$ photons cm$^{-2}$ s$^{-1}$ determined from the measured $j(NO_2)$ of $(6.43 \pm 0.30) \times 10^{-3}$ s$^{-1}$.

From the measured $j(NO_2)$, and with knowledge of $\sigma_\lambda$ and $\phi_\lambda$ for $NO_2$, the flux of the lamp was
determined to be $(1.63 \pm 0.09) \times 10^{16}$ photons cm$^{-2}$ s$^{-1}$ integrated over the 290 – 400 nm
wavelength range of the lamp. Using this flux, and the known $\sigma_\lambda$ and $\phi_\lambda$ for HONO over the
same wavelength range, $j(HONO)$ was determined to be $(1.66 \pm 0.10) \times 10^{-3}$ s$^{-1}$.
In the presence of aerosols under illuminated conditions, the rate of heterogeneous removal of
$NO_2$ at the aerosol surface to generate HONO is given by:

$$-\frac{d[NO_2]}{dt} = k[NO_2] \qquad (5)$$

where $k$ is the pseudo-first order rate coefficient for loss of $NO_2$ at the aerosol surface, and
which leads to the generation of HONO. The postulated mechanism for HONO production
from $NO_2$ is discussed in section 3.3.2 below, but for the definition of $k$ it is assumed to be a
first order process for $NO_2$. Integration of equation ((**5**) gives:

$$k = -\frac{\ln(\frac{[NO_2]_0 - [HONO]_t}{[NO_2]_0})}{t} \qquad (6)$$



where $[NO_2]_0 - [HONO]_t$ is the concentration of $NO_2$ at time $t$, assuming that each $NO_2$
molecule is quantitatively converted to a HONO molecule following surface uptake (see
section 3.3.2 for the proposed mechanism), and $[NO_2]_0$ is the initial concentration of $NO_2$.
Hence $k$ can be determined from equation ((**6**) using the measurement of the concentration of
HONO, [HONO], that has been generated from $TiO_2$ aerosol surfaces for an illumination time
of $t$ (and after subtraction of any background HONO produced from other sources and after
correction for loss via photolysis, see above), and with knowledge of $[NO_2]_0$.
The reactive uptake coefficient of $NO_2$ to generate HONO, $\gamma_{NO_2 \to HONO}$, defined as the
probability that upon collision of $NO_2$ with the $TiO_2$ aerosol surface a gas-phase HONO
molecule is generated, is given by:

$$\gamma_{NO_2 \to HONO} = \frac{4 \times k}{v \times SA} \qquad (7)$$

where $v$ is the mean thermal velocity of $NO_2$, given by $v = \sqrt{(8RT/(\pi M)}$ with $R$, $T$ and $M$ as
the gas constant, the absolute temperature and the molar mass of $NO_2$, respectively, SA is the
aerosol surface area density ($m^2 \, m^{-3}$) and $k$ is defined as above. Rearrangement of equation ((**7**)
gives:

$$k = \frac{\gamma_{NO_2 \to HONO} \times SA \times v}{4} \qquad (8)$$

Figure 4 shows the variation of $k$, determined from equation ((**6**) above with $t = 52$ s
(illumination time in the flow tube), against aerosol surface area density, SA, for $[NO_2]_0 = 200$
ppb and RH=15%, from which the gradient using equation (8) yields $\gamma_{NO_2 \to HONO} = (2.17 \pm$
$0.09) \times 10^{-5}$.

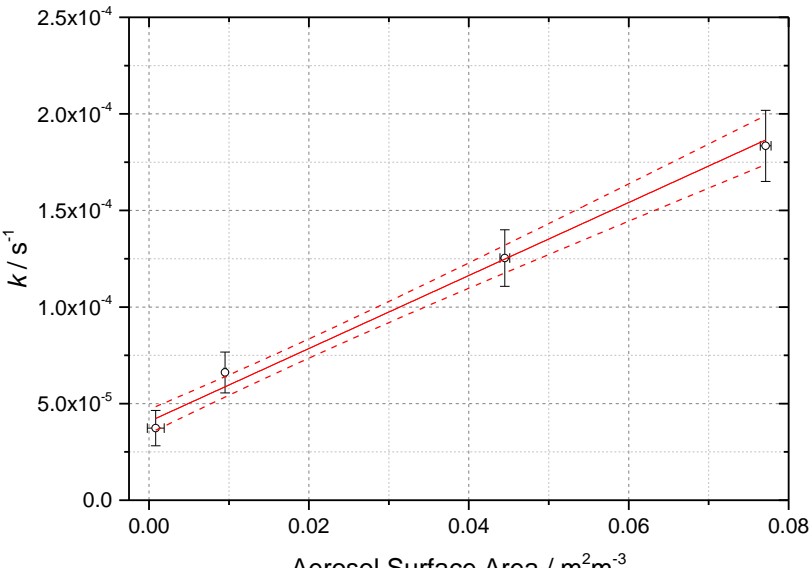

**Figure 4.** Pseudo-first-order rate coefficient for HONO production, $k$ (open circles) as a function of aerosol surface area for [$NO_2$]=200 ppb and RH=15 ± 1 %, T = 293 ± 3 K and a photolysis time of 52 ± 2 seconds. The red line is a linear-least squared fit including 1σ confidence bands (dashed lines) weighted to both $x$ and $y$ errors (1σ), the gradient of which yields $\gamma_{NO_2 \rightarrow HONO}$ = (2.17 ± 0.09)× $10^{-5}$, with the uncertainty representing (1σ). The total photon flux of the lamp (see Figure 2 for its spectral output) = (1.63 ± 0.09) × $10^{16}$ photons cm$^{-2}$ s$^{-1}$.

The uncertainty in $k$ (~20 %) shown in Figure 4 and determined by equation ((**6**) is mainly
controlled by the uncertainty in the HONO concentration (the HONO signal typically varies
between repeated runs for a given SA by ~ 10 % coupled with the 15 % error in calibration
factor), the initial $NO_2$ mixing ratio (10%), and the photolysis time, $t$ (~3 %). The uncertainty
in SA is determined by the uncertainty in the SMPS (15%). The error in the value of
$\gamma_{NO_2 \rightarrow HONO}$ (typically 20%) is calculated from the 1σ statistical error of the weighted fit shown
in Figure 4. An experiment performed using air yielded an uptake coefficient value within 7 %
of the equivalent experiment done in $N_2$, which is well within the experimental error.
**2.5  Box model description**
A kinetic scheme within the framework of a box model was used together with the differential
equation solver Facsimile 4.3.53 (MCPA software Ltd., 2020) to investigate the mechanism of
$NO_2$ adsorption on $TiO_2$ in the presence of light to produce HONO. The models were only
semi-explicit, focusing on determining the stoichiometric amounts of $NO_2$ needed to produce



a single HONO molecule in the gas-phase for comparison with the experimental dependence
of HONO production upon $NO_2$ mixing ratio, and to provide a predictive framework for
parameterising the HONO production rate with $NO_2$ mixing ratio in the atmosphere. Three
model scenarios were designed. The simplest model (Model 1) considered only the adsorption
of a single molecule of $NO_2$ to the $TiO_2$ surface, the surface conversion to HONO in the
presence of light and subsequent desorption of HONO, the latter assumed to occur rapidly. The
two further model scenarios investigated the effect of a 2:1 stoichiometric relationship between
the $NO_2$ adsorbed to the surface of $TiO_2$ and the HONO produced, via the formation of an $NO_2$
dimer. Model 2 incorporated an Eley-Rideal mechanism reliant on the adsorption of one $NO_2$
molecule to the surface followed by the subsequent adsorption of a second $NO_2$ molecule
directly onto the first (Figure 5). Model 3, however, features a Langmuir-Hinshelwood
mechanism of adsorption in which two $NO_2$ molecules adsorb to the surface, then diffuse to
one another before colliding on the surface and forming the *cis*-ONO-$NO_2$ dimer (Finlayson-
Pitts et al., 2003;de Jesus Madeiros and Pimentel, 2011;Liu and Goddard, 2012;Varner et al.,
2014). The formation of the asymmetric *cis*-ONO-$NO_2$ dimer followed by isomerisation to
form the asymmetric *trans*-ONO-$NO_2$ dimer has been suggested to have an enthalpic barrier
that is ~170 kJ mol$^{-1}$ lower than for direct isomerisation to *trans*-ONO-$NO_2$ from the symmetric
$N_2O_4$ dimer (Liu and Goddard, 2012). The dimerisation of $NO_2$ and subsequent isomerisation
to form *trans*-ONO-$NO_2$ has been suggested under dark conditions to lead to the formation of
both HONO and $HNO_3$ in the presence of water vapour (Finlayson-Pitts et al., 2003;de Jesus
Madeiros and Pimentel, 2011;Liu and Goddard, 2012;Varner et al., 2014). Although the
interaction of light with $TiO_2$ with the concomitant production of electron-hole pairs (R1) is
central to HONO formation, we do not specify here the exact mechanism by which the electron-
hole pairs interact with surface-bound species to generate HONO. We propose that the
interaction with light speeds up the autoionisation of *trans*-ONO-$NO_2$ to form $(NO^+)(NO_3^-)$,
which is represented by reactions R13 and R15 in Models 2 and 3 respectively. $(NO^+)(NO_3^-)$
can then react rapidly with surface adsorbed water leading to HONO formation (Varner et al.,

330    2014).

A schematic of the proposed mechanism investigated with Models 2 and 3 is shown in Figure
5, and consists of (i) the adsorption of $NO_2$ onto a surface site, (ii) the conversion of $NO_2$ to
form HONO via the formation of an $NO_2$ dimer intermediate on the surface via either a Eley-
Rideal or Langmiur Hinshelwood- type mechanism, (iii) subsequent desorption of HONO from
the surface, and finally (iv) competitive removal processes for HONO both on the surface and



in the gas-phase that are either dependent or independent on the $NO_2$ mixing ratio. The model
includes the gas-phase photolysis of $NO_2$ and HONO and the gas phase reactions of both
HONO and $NO_2$ with OH and $O(^3P)$ atoms.

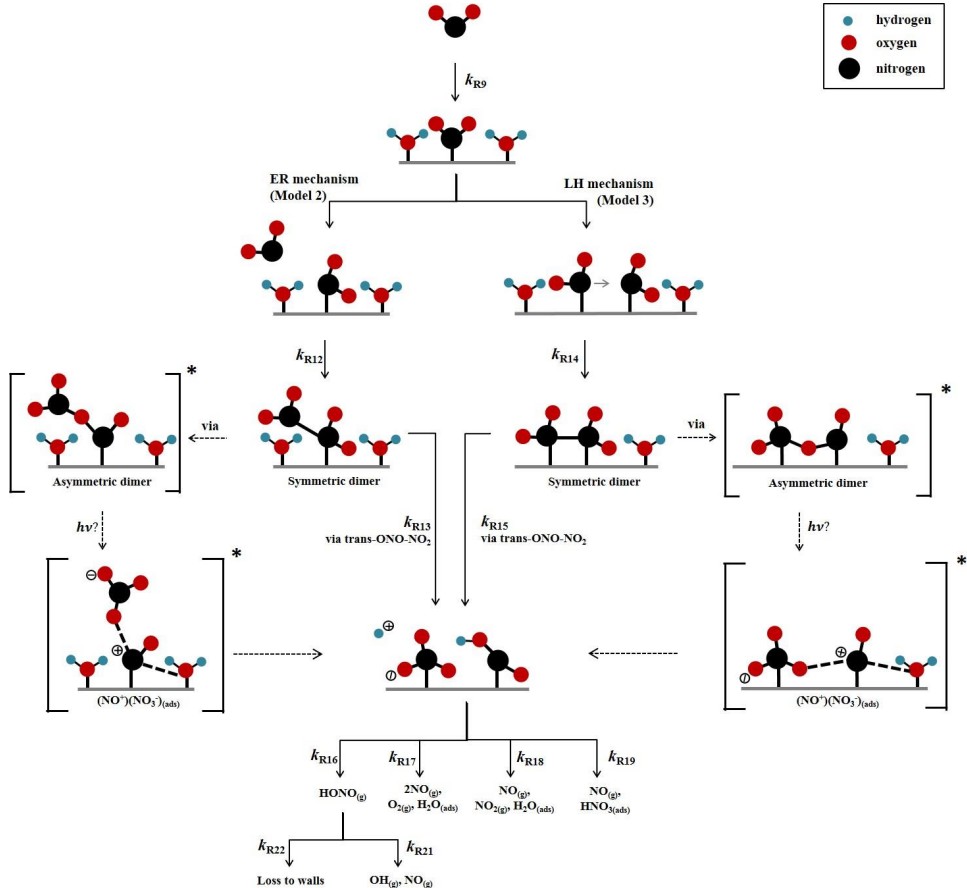

**Figure 5.** Schematic diagram of proposed mechanism of uptake of $NO_2$ on an aerosol surface in the presence
of water to form HONO. Both Eley Rideal, Model 2, and Langmuir Hinshelwood, Model 3, mechanisms are
shown with relevant estimated and calculated rate coefficients used in the models. $NO_2$ dependent and
independent loss reactions of HONO are also depicted. Nitrogen shown in black, oxygen shown in red and
hydrogen shown in blue. * denotes intermediate steps of the isomerisation of symmetric $N_2O_4$ to *trans*-ONO-
$NO_2$ which is then predicted to form HONO.

To the best of our knowledge the enthalpy of adsorption of $NO_2$ onto a $TiO_2$ surface has not
been determined, nor the bimolecular rate coefficients for the chemical steps on the surface
shown in Figure 5. Hence, for each of the steps a rate coefficient ($s^{-1}$ or $cm^3$ molecule$^{-1}$ $s^{-1}$) was
assigned, as given in Table 1, and with the exception of the experimentally determined $j(NO_2)$





and the calculated $j$(HONO), and the gas-phase rate coefficients which are known, the rate
coefficients were estimated, with the aim of reproducing the experimental $NO_2$ dependence of
the HONO production and $NO_2$ reactive uptake coefficient; justification of chosen values is
given below.

| Reactions | | Rate coefficient[d] |
|---|---|---|
| **Model 1** | | |
| R9 | $NO_{2(g)} + surface \rightarrow NO_{2(ads)}$ | $1\times10^{-3}$ |
| R10 | $NO_{2(ads)} \rightarrow HONO_{(ads)}$ | $1\times10^{-3}$ |
| R11 | $HONO_{(ads)} \rightarrow HONO_{(g)}$ | $1\times10^{-2}$ |
| **Model 2 and 3** | | |
| *Model 2 only – Eley-Rideal mechanism* | | |
| R12 | $NO_{2(g)} + NO_{2(ads)} \rightarrow NO_2 - NO_{2(ads)}$ | $1\times10^{-2}$ |
| R13 | $NO_2 - NO_{2(ads)} \xrightarrow{via\ trans-ONO-NO_2} HONO_{(ads)} + HNO_{3(ads)}$ | $5\times10^{-3}$ |
| *Model 3 only – Langmuir-Hinshelwood mechanism* | | |
| R14 | $NO_{2(ads)} + NO_{2(ads)} \rightarrow NO_{2(ads)} - NO_{2(ads)}$ | $1\times10^{-3}$ |
| R15 | $NO_{2(ads)} - NO_{2(ads)} \xrightarrow{via\ trans-ONO-NO_2} HONO_{(ads)} + HNO_{3(ads)}$ | $5\times10^{-3}$ |
| *Common to both Models 2 and 3* | | |
| R9 | $NO_{2(g)} + surface \rightarrow NO_{2(ads)}$ | $1\times10^{-1}$ |
| R16 | $HONO_{(ads)} \rightarrow HONO_{(g)}$ | $5\times10^{-2}$ |
| R17 | $HNO_{3(ads)} + HONO_{(ads)} \rightarrow 2NO_{(g)} + O_{2(g)} + H_2O_{(ads)}$ | $1\times10^{-3}$ |
| R18 | $HONO_{(ads)} + HONO_{(ads)} \rightarrow NO_{(g)} + NO_{2(g)} + H_2O_{(ads)}$ | $1\times10^{-3}$ |
| R19 | $NO_{2(g)}(or\ species\ such\ as\ NO_2^+) + HONO_{(ads)} \rightarrow NO_{(g)} + HNO_{3(ads)}$ | $5\times10^{-3}$ |
| R20 | $NO_{2(g)} + hv \rightarrow NO_{(g)} + O(^3P)_{(g)}$ | $6\times10^{-3a}$ |
| R21 | $HONO_{(g)} + hv \rightarrow OH_{(g)} + NO_{(g)}$ | $2\times10^{-3b}$ |
| R22 | $HONO_{(g)} \rightarrow wall\ loss$ | $1\times10^{-4}$ |
| R23 | $HONO_{(g)} + OH_{(g)} \rightarrow NO_{2(g)} + H_2O_{(g)}$ | $4.5\times10^{-12c}$ |
| R24 | $NO_{2(g)} + OH_{(g)} \xrightarrow{M} HNO_{3(g)}$ | $1\times10^{-11c}$ |
| R25 | $O(^3P)_{(g)} + NO_{2(g)} \rightarrow O_{2(g)} + NO_{(g)}$ | $1\times10^{-11c}$ |
| R26 | $O(^3P)_{(g)} + O_{2(g)} \xrightarrow{M} O_3$ | $1.5\times10^{-14c}$ |
| R27 | $O(^3P)_{(g)} + NO_{(g)} \xrightarrow{M} NO_{2(g)}$ | $1.7\times10^{-12c}$ |

**Table 1.** Reactions included in the chemical mechanism used to model $NO_2$ uptake onto $TiO_2$ aerosols. All rate
coefficients are estimated, as described in Section 2.5, with the exception of the $NO_2$ and HONO photolysis rate
coefficient and the gas phase rate coefficient which are known. [a]Measured using chemical actinometry with the
knowledge of the experimentally determined spectral output of the lamp and the cross-sections and quantum
yields of $NO_2$ and HONO, see section 2.4 for more detail. [b]Calculated using a photon flux of $(1.63 \pm 0.09) \times$



$10^{16}$ photons cm$^{-2}$ s$^{-1}$.$^{c}$(Sander et al., 2003). $^{d}$Rate coefficients are in the units of s$^{-1}$ for first-order processes or
cm$^3$ molecule$^{-1}$ s$^{-1}$ for second-order processes. $T$ for all k values is 298 K.
The modelled Gibbs free energy barrier for the isomerisation of $N_2O_4$ to form the asymmetric
ONO-NO$_2$ isomer (*cis* or *trans* conformation not specified)  was estimated by Pimental et al.,
(2007) to be 87 kJ mol$^{-1}$ with a rate coefficient as large as $2 \times 10^{-3}$ s$^{-1}$ in the aqueous phase at
298 K, stated in the study to confirm the Finlayson-Pitts model for the hydrolysis of NO$_2$ on
surfaces via the asymmetric *trans*-ONO-NO$_2$ dimer (Finlayson-Pitts et al., 2003). Using this
study as a guide, we estimated $k_{R13}$ and $k_{R15}$ as $5 \times 10^{-3}$ s$^{-1}$, slightly larger than that estimated
by Pimental et al., (2007) due to the presence of light. A study into the decomposition of HONO
on borosilicate glass surfaces suggested a rate coefficient for the loss HONO on the non-
conditioned chamber walls to be $(1.0 \pm 0.2) \times 10^{-4}$ s$^{-1}$ increasing to $(3.9 \pm 1.1) \times 10^{-4}$ s$^{-1}$ when
HNO$_3$ was present on the walls (Syomin and Finlayson-Pitts, 2003). From this we estimated a
light-accelerated loss rate coefficient of $1 \times 10^{-3}$ s$^{-1}$ for the loss of HONO$_{(ads)}$ by reaction with
itself, $k_{R18}$, and through reaction with HNO$_{3(ads)}$, $k_{R17}$. Both these reactions will occur on the
surface of the aerosol. We make the assumption that the rate of loss of HONO to the walls of
the chamber for this experiment is less than that of the heterogeneous loss reactions on the
photo-catalytic aerosol surface leading to a $k_{R22}$ of $1 \times 10^{-4}$ s$^{-1}$ as reported by (Syomin and
Finlayson-Pitts, 2003)**.** For $k_{R12}$-$k_{R15}$, initial values were adopted and were then adjusted to fit
the shape of the trend in experimental results of [HONO] and $\gamma_{NO_2 \rightarrow HONO}$ versus [NO$_2$],
discussed fully in Section 3.3.2. For completeness, gas-phase loss reactions of HONO and NO$_2$
with OH and the reactions of O($^3$P) with NO, NO$_2$ and O$_2$ were also included in the model,
R23-R27, though their inclusion had no effect on the HONO concentration. The rates of R23-
R27 within the model are much smaller than HONO loss reactions on the surface (R17-R19)
and the photolysis reactions (R21). For both Models 2 and 3, the adsorption of an NO$_2$ molecule
to the surface, $k_{R9}$, was assumed to be rapid and not the rate determining step. Likewise, the
desorption of HONO was also assumed to be rapid, faster than the loss rates of adsorbed HONO
but slower than the adsorption of NO$_2$; this was necessary for the model to reproduce the trend
in the experimental results of [HONO] versus [NO$_2$], discussed fully in Section 3.3.2.
**3   Results and Discussion**
**3.1   HONO production from TiO$_2$ aerosol surfaces in the presence of NO$_2$**
The production of HONO on TiO$_2$ aerosol surfaces was measured as a function of the initial
NO$_2$ mixing ratio. Figure 6 shows the dependence of the HONO concentration, measured at





384 the end of the flow tube, on the initial NO$_2$ mixing ratio for an aerosol surface area of (1.6 ±

385 0.8) × 10$^{-2}$ m$^2$ m$^{-3}$. A sharp increase in HONO production at a low mixing ratio of NO$_2$ was

386 seen followed by a more gradual reduction in HONO production after a peak production at ~

387 54 ± 5 ppb NO$_2$.

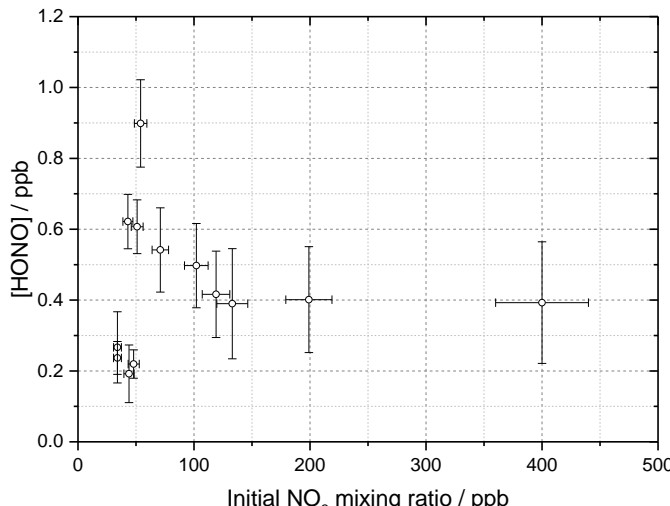

**Figure 6.** HONO concentration measured at the end of the flow tube as a function of the initial NO$_2$ mixing ratio, for the aerosol surface area density of (1.6 ± 0.8) × 10$^{-2}$ m$^2$ m$^{-3}$, relative humidity 15 ± 1 %, photon flux (1.63 ± 0.09) × 10$^{16}$ photons cm$^{-2}$ s$^{-1}$ (290-400 nm wavelength range), reaction time of 52 seconds and N$_2$ carrier gas. Each point is an average of up to 20 measurements at the same aerosol surface area and mixing ratio of NO$_2$. The highest concentration of HONO measured was 0.90 ± 0.12 ppb at [NO$_2$] = 54 ± 5 ppb. The *y* error bars represent 1σ while the *x* error bars represent the sum in quadrature of the errors in the N$_2$ and NO$_2$ gas flows and the NO$_2$ dilution. The SA varied over the experiments at different NO$_2$ mixing ratios leading to a larger error in the quoted SA.

388 Figure 7 shows the HONO concentration measured at the end of the flow tube over a range of

389 RH values for a fixed aerosol surface area density of (1.59 ± 0.16 × 10$^{-2}$ m$^2$ m$^{-3}$) and at two

390 NO$_2$ mixing ratios, displaying a peak in HONO production between 25 – 30 % RH. Above ~

391 37 % RH, for experiments including single-component TiO$_2$ aerosols, it was found that

392 significant aerosols were lost from the system before entering the flow tube, speculated to be

393 due to loss to the walls of the Teflon lines. As such the RH dependence was only studied up to





37 % RH, however a clear drop off in HONO production was seen for both $NO_2$ mixing ratios
studied after ~ 30 % RH.

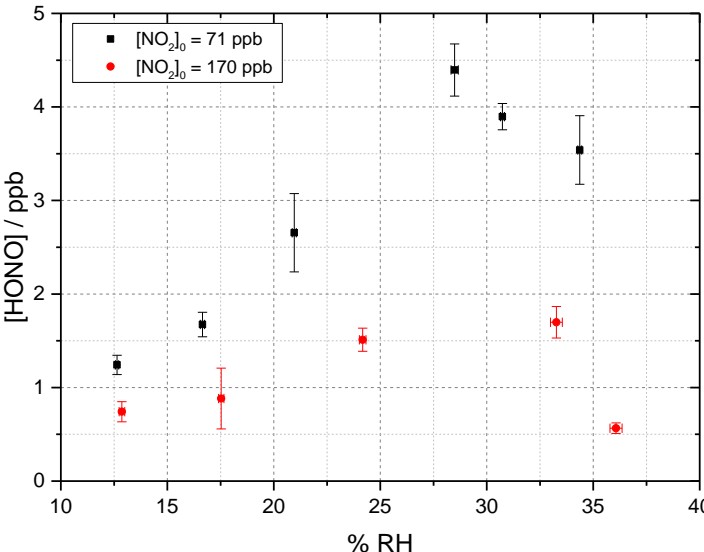

**Figure 7.** RH dependence of HONO production from illuminated $TiO_2$ aerosol surfaces at 295 K in $N_2$ at 71

(black) and 170 (red) ppb initial $NO_2$ mixing ratio. The aerosol surface area density was kept constant at $(1.59$

$\pm 0.16) \times 10^{-2}$ $m^2$ $m^{-3}$ with a photon flux of $(1.63 \pm 0.09) \times 10^{16}$ photons $cm^{-2}$ $s^{-1}$ and an illumination time of 52

$\pm 2$ seconds. The error bars represent $1\sigma$.

A dependence of HONO production upon RH was expected due to the potential role of water
as a proton donor in the production mechanism of HONO on $TiO_2$ surfaces (R2 and (R5, as
shown in Figure 5 (Dupart et al., 2014). The fractional surface coverage of water on the $TiO_2$
aerosol core, $V/V_m$, at 15 % RH and above was calculated using the parameterisation below,
which was determined using transmission IR spectroscopy (Goodman et al., 2001):

$$\frac{V}{V_m} = \left[\frac{c\left(\frac{P}{P_0}\right)}{1 - \left(\frac{P}{P_0}\right)}\right] \left[\frac{1 - (n+1)\left(\frac{P}{P_0}\right)^n + n\left(\frac{P}{P_0}\right)^{n+1}}{1 + (c-1)\left(\frac{P}{P_0}\right) - c\left(\frac{P}{P_0}\right)^{n+1}}\right] \tag{9}$$

where $V$ is the volume of water vapour adsorbed at equilibrium pressure $P$, $V_m$ is the volume
of gas necessary to cover the surface of $TiO_2$ particles with a complete monolayer, $P_0$ is the
saturation vapour pressure, $c$ is the temperature dependent constant related to the enthalpies of


adsorption of the first and higher layers (taken as 74.8 kJ mol$^{-1}$ for $TiO_2$ (Goodman et al., 2001))
and $n$ is the asymptotic limit of monolayers (8 for $TiO_2$ (Goodman et al., 2001)) at large values
of $P/P_0$.
At 15 % RH, a fractional water coverage of 1.09 was calculated to be present on the surface,
increasing to 1.50 at 35 % RH. It has been shown in previous work that HONO can be displaced
from a surface by water, leading to an increase in gas-phase HONO with RH (Syomin and
Finlayson-Pitts, 2003). The increase in HONO with RH to ~25-30 % RH could therefore be
attributed to both an increase in the concentration of the water reactant leading to more HONO
formation and the increase in displacement of HONO from the surface due to preferential
adsorption of water. A decrease in HONO production seems to occur above ~ 30 % RH, which
could be due to the increased water adsorption inhibiting either $NO_2$ adsorption or the
electron/hole transfer process (Gustafsson et al., 2006). $H_2O$ vapour adsorption is likely
enhanced by the superhydrophilic properties of $TiO_2$ surfaces under UV radiation meaning that
water monolayers form more quickly on the surface of $TiO_2$ owing to light-induced changes in
surface tension (Takeuchi et al., 2005;Gustafsson et al., 2006).
At the higher initial concentration of $NO_2$ = 170 ppb, the RH dependence showed a similar
peak in HONO production between ~25 - 30 % RH but less HONO was produced overall, as
expected from Figure 6 given the higher $NO_2$. Previous work on the production of HONO from
suspended $TiO_2$ aerosols reported a strong RH dependence of the uptake coefficient, $\gamma$, of $NO_2$
to form HONO with a peak at ~ 15 % RH and decreasing at larger RH (Gustafsson et al., 2006).
The same trend for the $NO_2$ uptake coefficient was observed by Dupart et al., 2014 on Arizona
test dust (ATD) aerosols with a peak in $\gamma$ at ~ 25 % RH. This increase in the RH at which the
uptake coefficient for $NO_2$ in going from $TiO_2$ to ATD aerosols was ascribed to the lower
concentration of $TiO_2$ present in ATD aerosols as opposed to single-component $TiO_2$ aerosols
used by Gustafsson et al., 2006 as well as by differences in particle size distribution. Gustafsson
et al., 2006 reported a larger aerosol size distribution with a bimodal trend with mode diameters
of ~ 80 and ~ 350 nm for single-component $TiO_2$ aerosols whereas Dupart et al., 2014 reported
a smaller unimodal aerosol size distribution for ATD aerosols with a mode diameter of ~110
nm. In this work we also see a larger aerosol size distribution, with a lower mode diameter of
~ 180 nm similar to Dupart et al., 2014 but for pure $TiO_2$ aerosols; aerosol size distribution
shown in Figure 2. Similar to the results of Dupart et al., 2014 we observe a trend inversion in
[HONO] vs RH at higher RH, between 25-30 %.



### 3.2 Dependence of reactive uptake coefficient on initial NO₂ mixing ratio


The reactive uptake coefficient, $\gamma_{NO_2 \rightarrow HONO}$ for NO₂→HONO on TiO₂ aerosol particles was
determined experimentally for 18 different initial NO₂ mixing ratios, and is shown in Figure 8.
For each initial NO₂ mixing ratio, the gradient of the first order rate coefficient for HONO
production, $k$, as a function of aerosol surface area density (e.g. Figure 4) and in conjunction
with equation ((8), was used to obtain $\gamma_{NO_2 \rightarrow HONO}$. The uptake coefficient initially increases
with NO₂, reaching a peak at $\gamma_{NO_2 \rightarrow HONO} = (1.26 \pm 0.17) \times 10^{-4}$ for an initial NO₂ mixing ratio
of $51 \pm 5$ ppb, before sharply decreasing as the NO₂ mixing ratio continues to increase above
this value.

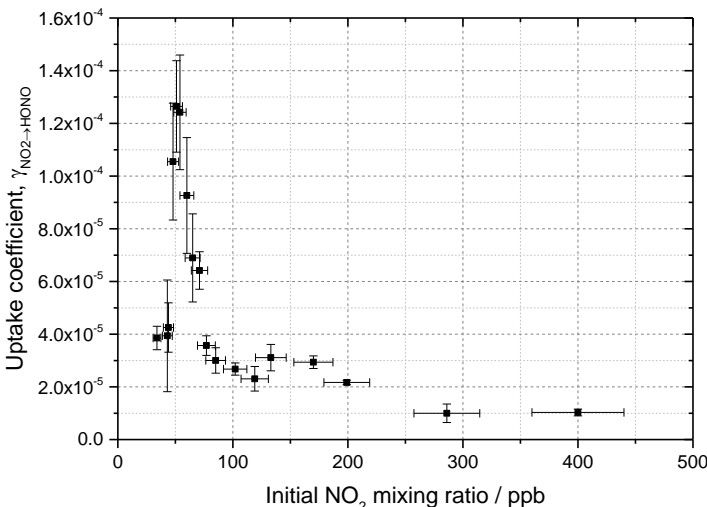

**Figure 8** Experimental results showing the reactive uptake coefficients of NO₂ to form HONO, $\gamma_{HONO \rightarrow NO_2}$ onto TiO₂ aerosol surfaces as a function of the initial NO₂ mixing ratio. All experiments were conducted in N₂ at 295 K at $15 \pm 1$ % RH, a photon flux of $(1.63 \pm 0.09) \times 10^{16}$ photons cm$^{-2}$ s$^{-1}$ and an illumination time of $52 \pm 2$ seconds. $\gamma_{HONO \rightarrow NO_2}$ was determined for each NO₂ mixing ratio from the gradient of the pseudo-first-order rate coefficient for HONO production, $k$, versus aerosol surface area density varied from 0 - 0.04 m² m$^{-3}$ (e.g. as shown in **Figure** 4) and Equation ((8).

The increase in uptake coefficient with NO₂ at low NO₂ (< 51 ppb) has not been seen previously
in studies of HONO production from TiO₂ containing aerosols with similar [NO₂] ranges
(Gustafsson et al., 2006;Ndour et al., 2008;Dupart et al., 2014) or other aerosol surfaces
(Bröske et al., 2003;Stemmler et al., 2007). It is worth noting that several of these studies



reported the overall uptake of $NO_2$ onto aerosol surfaces and not specifically the uptake to form
HONO, although HONO was indirectly measured in all studies noted here (Gustafsson et al.,
2006;Ndour et al., 2008;Dupart et al., 2014). For single-component $TiO_2$ aerosols, Gustafsson
et al., (2006) reported a uptake coefficient, $\gamma_{NO_2}$, of $9.6 \times 10^{-4}$ at 15 % RH and 100 ppb $NO_2$.
Taking into account the HONO yield of 0.75 given by (Gustafsson et al., 2006), an estimated
$\gamma_{NO_2 \rightarrow HONO} = 7.2 \times 10^{-4}$ is determined and can be compared to the value observed in this work
at 15 % RH and 100 ppb $NO_2$, ($\gamma_{NO_2 \rightarrow HONO} = (2.68 \pm 0.23) \times 10^{-5}$). The $\gamma_{NO_2 \rightarrow HONO}$ we
determine is 27 times smaller than reported by Gustafsson et al., (2006). This difference is
mostly due to the lower experimental photon flux in our setup, ~19 times less at $\lambda_{max} = 365$
nm owing to the use of one 15 W UV lamp to irradiate the flow tube (Boustead, 2019)
compared to Gustafsson et al., 2006 which utilised four 18 W UV lamps.
The origins of the increase in $\gamma_{NO_2 \rightarrow HONO}$, together with reaching a maximum and the
subsequent decrease at larger $NO_2$ mixing ratios was investigated using the kinetic box model
and postulated mechanism for HONO production described in Section 2.5. The aim was to
compare the observed production of HONO and $\gamma_{NO_2 \rightarrow HONO}$ with the modelled values, as a
function of $NO_2$ mixing ratio. The skill of the model to reproduce the observed behaviour
enables a validation of the postulated mechanism for HONO production, and variation of the
kinetic parameters enables the controlling influence of different steps in the mechanism on
HONO production to be evaluated.

### 468    3.3   Modelling the HONO production mechanism on illuminated $TiO_2$
### 469       aerosol surfaces

The HONO production on illuminated $TiO_2$ aerosol surfaces was investigated for each of the
mechanisms outlined in Table 1.

### 472    3.3.1   Model 1

Model 1 (see Table 1 and Figure 5), which contains the simplest mechanism, was designed to
reproduce the decreasing value of the $NO_2$ uptake coefficient to form HONO, $\gamma_{NO_2 \rightarrow HONO}$, with
increasing $NO_2$ and also the plateauing at higher $NO_2$ mixing ratios caused by $NO_2$ reaching a
maximum surface coverage, as seen by Stemmler et al., (2007). A decrease in the uptake
coefficient of $NO_2$, $\gamma_{NO_2}$, onto dust aerosol surfaces was also seen in studies where the
formation of HONO from $NO_2$ uptake was not directly studied (Ndour et al., 2008;Dupart et
al., 2014). The mechanism for Model 1 which is given in Table 1 describes the adsorption of



one $NO_2$ molecule to a surface site which then undergoes the reaction which forms HONO,
followed by desorption of HONO to the gas-phase, R9-R11. Any representation of the specific
chemical processes which convert $NO_2$ to HONO on the surface following the initial photo-
production of electron-hole pairs in the $TiO_2$ structure (R2) was not included here as the
primary focus was to produce the relationship between $\gamma_{NO_2 \rightarrow HONO}$ and the $NO_2$ mixing ratio.
Gustafsson et al., (2006) reported that the measured rate of photo-induced HONO production
is 75% that of the rate of $NO_2$ removal, whereas the dark disproportionation reaction (**R28**)
would predict a 50% yield, and hence that the HONO observed in their studies is not simply a
photo-enhancement of:

$$2NO_{2(ads)} + H_2O_{(ads)} \rightarrow HONO_{(g)} + HNO_{3(ads)} \qquad \text{(R28)}$$

Gustafsson et al., (2006) suggests that an oxidant on the surface is produced following the
creation of the electron-hole pair (OH is generated in (R2)), and suggests $H_2O_2$ as a possibility,
which is consistent with the observation of OH and $HO_2$ radicals produced from the surface of
illuminated $TiO_2$ aerosols (Moon et al., 2019). For Model 1, outputs for the predicted
concentration of HONO and the reactive uptake coefficient, $\gamma_{NO_2 \rightarrow HONO}$, as a function of initial
$NO_2$ mixing ratio are shown in Figure 9.

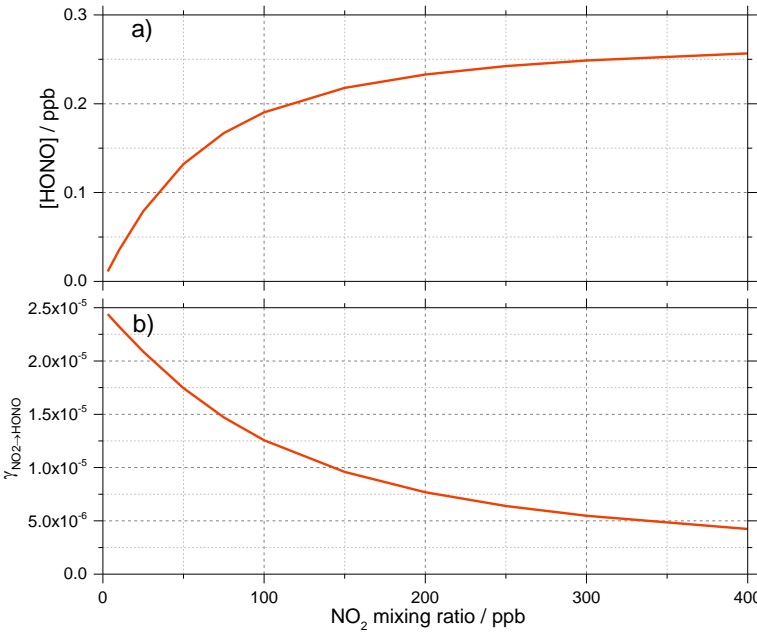





**Figure 9** Model 1 calculations for (a) the concentration of HONO and (b) the reactive uptake coefficient to form HONO, $\gamma_{NO_2 \rightarrow HONO}$, as a function of $NO_2$ mixing ratio for a model run time of 52 s. The estimated rate coefficients used in this model are shown in Table 1.

For a run time of 52 s, equal to that of the experimental illumination time, Model 1 predicts an
increase in HONO production with increasing $NO_2$ mixing ratio until the HONO concentration
begins to plateau, reaching ~0.25 ppb at $[NO_2]$ = 400 ppb, presumably owing to saturation on
active aerosol surface sites by $NO_2$. This leads to the modelled reactive uptake coefficient,
$\gamma_{NO_2 \rightarrow HONO}$, monotonically decreasing with increasing $NO_2$ mixing ratio; a variation in $NO_2$
uptake coefficient similar to that seen in previous photo-enhanced $NO_2$ aerosol uptake studies
(Bröske et al., 2003;Stemmler et al., 2007;Ndour et al., 2008;Dupart et al., 2014). However,
the model predictions for Model 1 do not reproduce the experimental variations shown in
Figure 6 and Figure 8, in which there is an observed initial rise and then a fall in both the
HONO concentration and $\gamma_{NO_2 \rightarrow HONO}$ with increasing $NO_2$ mixing ratio. Hence, additional
processes were considered in the model in order to try to reproduce this behaviour.
**3.3.2   Models 2 and 3. Investigating the role of $NO_2$ dimerisation for the surface**
**formation of HONO, and including additional surface losses of HONO**
As the experimental $\gamma_{NO_2 \rightarrow HONO}$ increases with $NO_2$ at low $NO_2$ (Figure 8), we postulate in
Models 2 and 3 that the production of HONO under illuminated conditions is not fully first
order in $NO_2$ and requires more than one $NO_2$ molecule to form HONO, consistent with the
formation of the symmetric $NO_2$ dimer ($N_2O_4$) followed by isomerisation on the surface to
form the asymmetric *trans*-ONO-$NO_2$ dimer, which has been suggested to be more reactive
with water than the symmetric $N_2O_4$ dimer (Finlayson-Pitts et al., 2003;Ramazan et al.,
2004;Ramazan et al., 2006;Liu and Goddard, 2012) due to the autoionisation to form
$(NO^+)(NO_3^-)$ which we propose is accelerated by the presence of light; the full mechanism for
which is shown in Figure 5. A recent rotational spectroscopy study found that the *trans*-ONO-
$NO_2$ was better described as the ion pair $(NO^+)(NO_3^-)$ (Seifert et al., 2017). Reaction of the
$(NO^+)(NO_3^-)$ ion pair with surface adsorbed water can then lead to the formation of HONO and
$HNO_3$, the feasibility of which is supported by molecular dynamics simulation studies (Varner
et al., 2014). While the symmetric $N_2O_4$ dimer is favoured as it is the most stable conformer,
the asymmetric forms have been experimentally observed in several studies (Fateley et al.,
1959;Givan and Loewenschuss, 1989b, a, 1991;Pinnick et al., 1992;Forney et al., 1993;Wang
and Koel, 1998, 1999;Beckers et al., 2010). A more recent *ab initio* study of $NO_2$ adsorption





at the air-water interface suggested an orientational preference of $NO_2$ on the surface, with both
oxygen atoms facing away from the interface which may imply that the asymmetric dimer
$ONO-NO_2$ can form directly, meaning the high barrier between the symmetric and asymmetric
forms does not need to be overcome (Murdachaew et al., 2013).
The energy barrier to isomerisation of symmetric $N_2O_4$ in the gas-phase may be reduced due
to the interaction with water adsorbed on surfaces. We therefore rule out the dimer in the gas-
phase adsorbing onto the surface first, and then reacting to form HONO (Varner et al., 2014).
An interesting question is whether the first $NO_2$ molecule adsorbed to the surface dimerises via
the addition of a gaseous $NO_2$ via an Eley-Rideal (ER) type process, or whether a Langmuir-
Hinshelwood (LH) type mechanism is operating in which both $NO_2$ molecules are first
adsorbed and then diffuse together on the surface forming $N_2O_4$. Both ER and LH mechanisms
to form the $NO_2$ dimer have been included in the model, denoted as Model 2 and Model 3,
respectively. The outputs for Models 2 and 3 (see Table 1 for details of the processes included)
for the HONO concentration and $\gamma_{NO_2 \rightarrow HONO}$ as a function of $NO_2$ are shown in Figure 10
together with the experimental data. The stoichiometric relationship of the requirement of two
$NO_2$ molecules forming HONO on the surface was key to reproducing the experimental trend
of first an increase and then a decrease in both the HONO concentration and the reactive uptake
coefficient with the initial $NO_2$ mixing ratio.

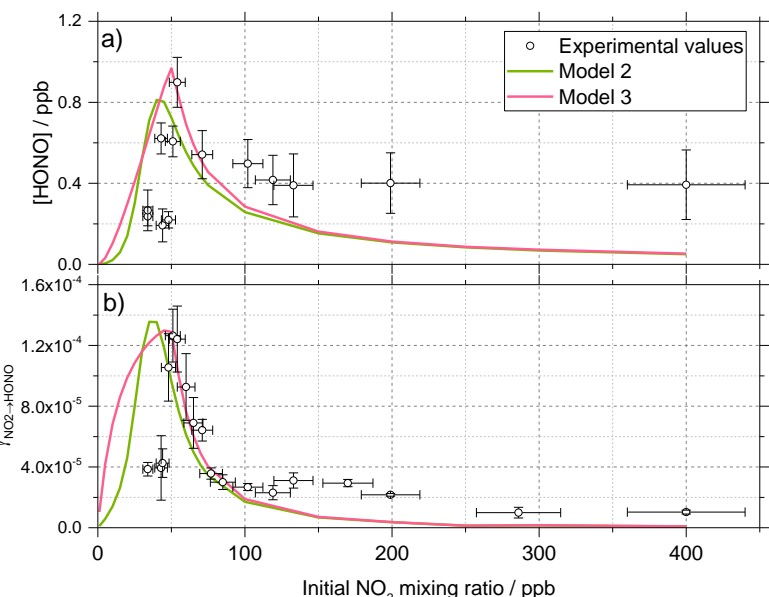

**Figure 10.** Experimental values (open circles with 1σ error bars), Model 2 (green line) and Model 3 (pink line) calculations for (a) HONO concentration after 52 s illumination and (b) NO$_2$ reactive uptake coefficient, $\gamma_{NO_2 \rightarrow HONO}$, as a function of the initial NO$_2$ mixing ratio. The mechanisms used for these model runs included a 2:1 stoichiometric relationship between the NO$_2$ adsorbed on the TiO$_2$ aerosol surface and the HONO produced, as well as additional HONO loss reactions which are dependent on NO$_2$, see Table 1 for details. Models 2 and use an Eley-Rideal and Langmuir-Hinshelwood mechanisms, respectively, for the formation of the NO$_2$ dimer on the aerosol surface.

In previous work that investigated HONO production from humic acid aerosols, a saturation
effect was seen with HONO production plateauing with increasing NO$_2$ mixing ratio (Stemmler
et al., 2007), with the decreasing uptake coefficient, $\gamma_{NO_2 \rightarrow HONO}$, with increasing NO$_2$ being
attributed to NO$_2$ fully saturating available surface sites. However, the observed decrease of
[HONO] at the high NO$_2$ mixing ratios shown in Figure 8 and Figure 10a suggests that
additional reactions on the surface may remove HONO and result in the reduction of [HONO]
that is measured. As [HONO] decreases with the increase in the NO$_2$ mixing ratio, the removal
process should either involve NO$_2$ directly:

$$HONO + NO_2 \rightarrow NO + HNO_3 \qquad (R19)$$

or involve species made rapidly from NO$_2$ on the surface, such as NO$_2^+$:



$$HONO_{(ads)} + NO_{2\ (ads)}^{+} \rightarrow H^{+} + 2NO + O_2 \tag{R19a}$$

which may be present at high enough concentrations of HNO$_3$ on the surface (Syomin and
Finlayson-Pitts, 2003) or following reaction with h$_{VB}^{+}$, or a product of the reaction of
$O_2^{-}$ (or $e_{CB}^{-}$) with NO$_2$ (R4) i.e. NO$_2^{-}$. Transition state theory (TST) studies of the gas-phase
reaction of HONO with NO$_2$ to form HNO$_3$ calculated a large activation energy which varied
depending on whether the reaction occurs via O abstraction by HONO (159 kJ mol$^{-1}$) or via
OH abstraction via NO$_2$ (~133-246 kJ mol$^{-1}$)(Lu et al., 2000). In the gas-phase these reactions
are too slow to be important but they could be enhanced on the surface, potentially more so on
a photoactive surface such as TiO$_2$. The NO$_2$ dependent loss reaction, $k_{R19}$ in Table 1, was
necessary in the model to reproduce the sharp decrease in [HONO] versus NO$_2$ seen
experimentally after ~54 ppb NO$_2$. Without $k_{R19}$ the modelled [HONO] continued to increase
to a plateau, as seen in Model 1 (see Figure 9). In order to observe the model output seen in
Figure 10 for model 2 and 3, $k_{R19}$ also had to be slower than the desorption of HONO from the
surface, $k_{R16}$.
The addition of an NO$_2$ dependent loss reaction to both Model 2 and 3 had the most significant
effect on the trend in modelled HONO concentration. Though it is also possible that a
secondary product could remain adsorbed and therefore block active sites on the TiO$_2$ surface,
effectively poisoning the photo-catalyst, NO$_2$ independent loss reactions in the model, $k_{R17}$ and
$k_{R18}$ had little effect on the trend in [HONO] vs NO$_2$, only having an effect on the overall
[HONO]. HNO$_3$ has however been shown to remain adsorbed to surfaces once formed
(Sakamaki et al., 1983;Pitts et al., 1984;Finlayson-Pitts et al., 2003;Ramazan et al., 2004) and
may also react with adsorbed HONO, further reducing the product yield (Finlayson-Pitts et al.,
2003): these NO$_2$ independent loss reactions may therefore become more important at higher
NO$_2$ concentrations and hence surface concentrations of HONO and HNO$_3$:

$$HONO_{(ads)} + HNO_{3(ads)} \rightarrow 2NO_{(g)} + H_2O_{(ads)} + O_{2(g)} \tag{R17}$$

The photolysis of particulate nitrate was not considered in Models 2 or 3, due to the lack of
particulate nitrate in the system at $t$=0. The gas-to-particle conversion of any HNO$_3$ formed
was not considered to be important due to the assumption that most HNO$_3$ formed would
remain adsorbed to the aerosol surface (Sakamaki et al., 1983;Pitts et al., 1984;Finlayson-Pitts
et al., 2003;Ramazan et al., 2004).





For Model 2, which includes the production of HONO via the Eley-Rideal mechanism, in order
to reproduce the experimentally observed sharp increase followed by a decrease in both
[HONO] and $\gamma_{NO_2 \to HONO}$ as a function of increasing $NO_2$ mixing ratio, the modelled rate
coefficient for the adsorption of a gas-phase $NO_2$ molecule to another the surface adsorbed
$NO_2$ to initially form the symmetric $N_2O_4$ dimer, $k_{R12}$, had to be larger than for the isomerisation
step to form HONO and $HNO_3$ via *trans*-ONO-$NO_2$, $k_{R13}$. Interestingly, for HONO production
via the Langmuir-Hinshelwood mechanism, Model 3, the modelled rate coefficient for the
diffusion of one $NO_2$ molecule across the surface to form the dimer with another $NO_2$ molecule,
$k_{R14}$, had to be smaller than for the isomerisation step, $k_{R15}$, to more closely represent the
experimental results for the uptake coefficient. Additionally, in order to reproduce the
experimental trend in HONO formation as a function of $NO_2$ mixing ratio, the rate coefficient
for the $NO_2$ dependent loss reaction, $k_{R19}$, had to be larger than the $NO_2$ independent reactions,
$k_{R17}$ and $k_{R18}$, leading to $k_{R19} = 5 \times 10^{-3}$ s$^{-1}$. The modelled HONO concentration also sensitive
to the active site surface concentration: Model 3 required an active site surface concentration
2.5 times that of Model 2 to reproduce the peak in [HONO] at ~ 51 ppb $NO_2$ observed in the
experimental results. The reason for this is due to the difference in active site occupation in the
2 models: one active site is being occupied by two $NO_2$ molecules per HONO formed in Model
2 as opposed to Model 3 where two active sites are occupied per HONO formed. Regardless
of the choice of an Eley Rideal or Langmuir Hinshelwood mechanism, both models reproduce
the general shape of [HONO] and $\gamma_{NO_2 \to HONO}$ with $NO_2$, providing evidence that two $NO_2$
molecules are required to form HONO.
## 3.4 HONO production from illumination of a mixed NH₄NO₃/TiO₂ aerosol
## in the absence of NO₂
The photolysis of particulate nitrate has been postulated as a source of HONO under ambient
sunlit conditions during several field campaigns, from both aircraft and ground based
measurements (Reed et al., 2017;Ye et al., 2017a;Ye et al., 2017b). Here, experiments were
carried out to investigate the formation of HONO from particulate nitrate photolysis, with and
without the addition of a photo-catalyst. This is of significant interest for marine environments
downwind of arid desert regions due to the availability of $TiO_2$ or other photocatalytic materials
within aerosols in dust plumes that are transported from these regions (Hanisch and Crowley,

609    2003).





Using the aerosol flow tube setup described in Sections 2.1-2.4, an aqueous solution of
ammonium nitrate (5 g NH$_4$NO$_3$ in 500 ml milli-Q water) was used to generate nitrate aerosols.
At the RH used in this experiment, ~ 50 %, the aerosols were still deliquesced. For these
experiments the residence time of the aerosols in the illuminated region of the flow tube was
30 seconds (flow rate ~ 6 lpm), with the production of HONO following illumination measured
as a function of aerosol surface area density. The number of lamps was increased from 1 to 4,
increasing the photon flux from $(1.63 \pm 0.09) \times 10^{16}$ to $(8.21 \pm 2.39) \times 10^{16}$ photons cm$^{-2}$ s$^{-1}$
and $j$(NO$_2$) from $(6.43 \pm 0.30) \times 10^{-3}$ to $(3.23 \pm 0.92) \times 10^{-2}$ s$^{-1}$. The $j$(NO$_2$), $j$(HONO) and flux
values for 4 lamps were more than 4 times that of 1 lamp only due to the lamp casings being
mirrored, and so with 4 lamps, with 2 lamps on either side of the flow tube, the casings reflected
the light back into the flow tube, increasing the effective light intensity. For these experiments,
no gaseous NO$_2$ was added to the gas entering the flow tube. As shown in Figure 11, for the
illumination of pure nitrate aerosols, although a small amount of HONO was observed at higher
aerosol loadings, no statistically significant production of HONO was seen.

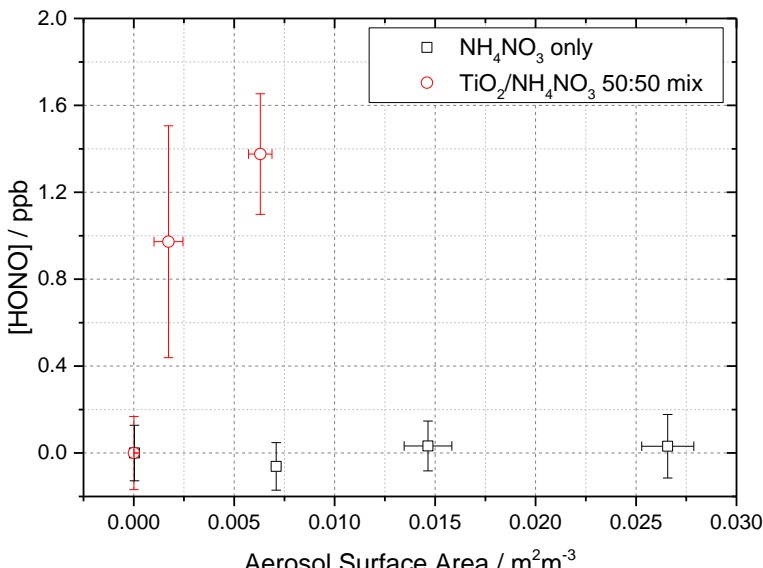

**Figure 11.** Dependence of the HONO concentration generated as a function of aerosol surface area density for pure NH$_4$NO$_3$ aerosol (black open squares, error bars represent 1$\sigma$) and 1:1 TiO$_2$/NH$_4$NO$_3$ mixed aerosol (red open circles, error bars represent 1$\sigma$). Both experiments were performed in N$_2$ at 295 K, an illuminated residence time of 30 s, and a lamp photon flux of $(8.29 \pm 2.39) \times 10^{16}$ photons cm$^{-2}$ s$^{-1}$. The NH$_4$NO$_3$ only experiment was performed at ~50 ± 5 % RH while the TiO$_2$/NH$_4$NO$_3$ mix experiment was performed at 20 ± 2





% RH. For all points, the background HONO seen observed without illumination has been subtracted. At zero aerosol surface area density there is no HONO generated from the walls of the flow tube.

A second set of experiments were performed with an aqueous solution of titanium dioxide and
ammonium nitrate combined in a 1:1 mass ratio to give a $TiO_2/NH_4NO_3$ aerosol mixture (5 g
$NH_4NO_3$ and 5 g $TiO_2$ in 500 ml milli-Q water) to investigate if the photo-catalytic properties
of $TiO_2$ facilitate the production of HONO in the presence of nitrate. The RH was decreased to
ensure the maximum $TiO_2$ photocatalytic activity (Jeong et al., 2013). A recent study using
Raman micro spectroscopy to observe phase changes in salt particles reported an efflorescence
point of pure ammonium nitrate to be between 13.7-23.9 % RH (Wu et al., 2019). It is possible
therefore that at the RH used in this experiment, ~ 20 %, the aerosols were still deliquesced.
As shown in Figure 11, the presence of $TiO_2$ in the aerosol mixture showed a significant
production of HONO without the presence of $NO_2$, a potentially significant result for the
production of HONO in low $NO_x$ environments in the presence of mixed dust/nitrate aerosols,
for example in oceanic regions off the coast of West Africa, or in continental regions impacted
by outflow from the Gobi desert. Using the Aerosol Inorganic Model (AIM) (Clegg et al.,
1998;Wexler and Clegg, 2002), the nitrate content of the aerosol at 50 % RH was calculated.
From this and the aerosol volume distribution given by the SMPS, the $[NO_3^-]$ within the
aerosols could be calculated. The formation of HONO by photolysis of particulate nitrate is
given by:

$$\frac{d[\text{HONO}]}{dt} = j(\text{pNO}_3)[\text{NO}_3^-] \qquad (10)$$

and hence:

$$[\text{HONO}] = j(\text{pNO}_3)[\text{NO}_3^-]t \qquad (11)$$

where $j(\text{pNO}_3)$ is the photolysis frequency of nitrate for the lamps used in these experiments
and $t$ is the illumination time of the experiment. With knowledge of [HONO], $[NO_3^-]$ and $t =$
30 s, $j(\text{pNO}_3)$ can be calculated from a measurement of [HONO] as a function of $[NO_3^-]$, as
shown in Figure 12, for the mixed nitrate/ $TiO_2$ experiment.

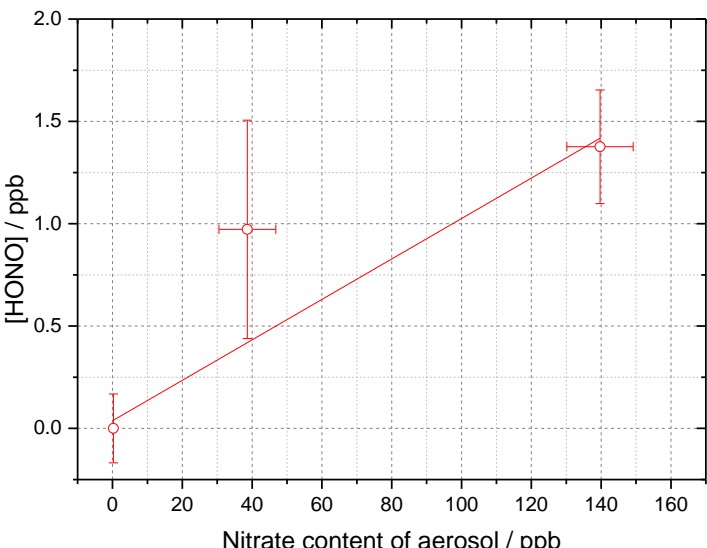

**Figure 12**. Dependence of [HONO] on the calculated nitrate concentration in the aerosol (using the AIM model) for the mixed $TiO_2$/ammonium nitrate aerosol experiment. Using Equation (**10** and for $t = 30$ s, the gradient gives $j(pNO_3) = (3.29 \pm 0.89) \times 10^{-4}$ s$^{-1}$. Experiment performed at $15 \pm 1$ % RH, in $N_2$ at 295 K with a lamp photon flux of $(8.29 \pm 2.39) \times 10^{16}$ photons cm$^{-2}$ s$^{-1}$. For all points, the background HONO seen observed without illumination has been subtracted.

When using the 4 lamps together, the experimental particulate nitrate photolysis rate, $j(pNO_3)$,
was determined to be $(3.29 \pm 0.89) \times 10^{-4}$ s$^{-1}$ for the mixed nitrate/$TiO_2$ aerosol. From this, it
is possible to estimate $j(pNO_3)$ for ambient conditions typical of the tropical marine boundary
layer. Taking the ratio of the experimental $j(HONO)$ for 4 lamps $((8.35 \pm 0.18) \times 10^{-3}$ s$^{-1})$ and
the measured $j(HONO)$ from the RHaMBLe campaign held at the Cape Verde Atmospheric
Observatory [May-June,2007] $(1.2 \times 10^{-3}$ s$^{-1})$ (Carpenter et al., 2010;Whalley et al., 2010;Reed
et al., 2017) and assuming that $j(pNO_3)$ and $j(HONO)$ scale in the same way, ambient $j(pNO_3)$
can be determined from:

$$j(pNO_3)_N = j(pNO_3) \times \frac{1.2 \times 10^{-3}}{j(HONO)} \qquad (12)$$

where $j(pNO_3)_N$ is the photolysis rate coefficient of particulate nitrate at Cape Verde, $j(pNO_3)$
is the experimentally determined photolysis rate coefficient of particulate nitrate to form


HONO and $j$(HONO) is the HONO photolysis rate coefficient calculated from the
experimentally determined $j$(NO$_2$).
Using $j$(pNO$_3$) = (3.29 ± 0.89) × 10$^{-4}$ s$^{-1}$, the rate of HONO production from nitrate photolysis
at Cape Verde was calculated to be $j$(pNO$_3$)$_N$= (4.73 ± 1.01) × 10$^{-5}$ s$^{-1}$ from the mixed
nitrate/TiO$_2$ aerosol experiment. Although for pure nitrate aerosol in the absence of TiO$_2$ the
data were scattered and the HONO production small (Figure 11), an upper limit estimate of
$j$(pNO$_3$)$_N$=(1.06 ± 1.15) × 10$^{-6}$ s$^{-1}$ under conditions at Cape Verde could be made using
equation (11), as done for rate of HONO production from mixed nitrate/TiO$_2$ aerosols. The
atmospheric implications of this will be considered below.

## 665    4    Implications of HONO production from TiO$_2$ for tropospheric chemistry

### 666    4.1    Production of HONO from sunlight aerosols containing TiO$_2$ in the
### 667            presence of NO$_2$

For the reactive uptake of NO$_2$ onto illuminated TiO$_2$ particles as a function of the initial NO$_2$
mixing ratio, as shown in Figure 8, a maximum value of $\gamma_{NO_2 \rightarrow HONO}$ = (1.26 ± 0.17) × 10$^{-4}$
was determined at 51 ± 5 ppb NO$_2$ for a photon flux from the lamp of (1.63 ± 0.09) × 10$^{16}$
photons cm$^{-2}$ s$^{-1}$. These experiments were for single-component TiO$_2$ particles, and so for dust
aerosols a value of $\gamma_{NO_2 \rightarrow HONO}$ = (1.26 ± 0.17) × 10$^{-5}$ is appropriate assuming a 10 % fraction
of TiO$_2$ and/or other photoactive materials (which behave similarly for HONO production) in
mineral dust (Hanisch and Crowley, 2003). Dust aerosols are transported from the Gobi desert
to urban areas of China where high NO$_x$ and nitrate aerosol concentrations have been observed
and in these areas HONO production facilitated by photo-catalysts may be important (Saliba
et al., 2014).
Using an average daytime maximum for [NO$_2$], $j$(NO$_2$) and aerosol surface area measurements
for a non-haze period in May-June in 2018 in Beijing, of 50 ppb, 1 × 10$^{-2}$ s$^{-1}$ and 2.5 × 10$^{-3}$ m$^2$
m$^{-3}$ (of which a maximum of 0.3 % was assumed to be TiO$_2$, though this could be higher in
dust impacted events (Schleicher et al., 2010)) respectively, a production rate of HONO of 1.70
× 10$^5$ molecules cm$^{-3}$ s$^{-1}$ (~24.8 ppt h$^{-1}$) has been estimated using the maximum reactive uptake
coefficient measured in this work, $\gamma_{NO_2 \rightarrow HONO}$ = (1.26 ± 0.17) × 10$^{-4}$. The lamp used to
illuminate the TiO$_2$ aerosols in these experiments gives rise to $j$(NO$_2$) = (6.43 ± 0.3) × 10$^{-3}$ s$^{-1}$,
and so $\gamma_{NO_2 \rightarrow HONO}$ has been scaled by a factor of 1.55 to match the noon $j$(NO$_2$) measured in
May-June 2018 in Beijing (10$^{-2}$ s$^{-1}$), to take into account the relatively small difference in



experimental and atmospheric photon flux for Beijing. The HONO production rate estimated
here for noontime summer [May-June 2018] in Beijing (~25 ppt hr$^{-1}$) is similar to the value for
the maximum production of HONO from urban humic acid aerosol surfaces in Europe, 17 ppt
h$^{-1}$ at 20 ppb $NO_2$ reported by Stemmler et al., 2007. For comparison, the net gaseous
production rate of HONO at noon in May-June 2018 Beijing was determined from the
measured rate of gas-phase production and losses:

$$P_{HONO} = k_{OH+NO}[\text{OH}][\text{NO}] - (\,j(\text{HONO}) \times [\text{HONO}] + k_{OH+HONO}[\text{OH}][\text{HONO}]) \quad (13)$$

where $k_{OH+NO} = 3.3 \times 10^{-11}$ cm$^3$ molecule$^{-1}$ s$^{-1}$ (Atkinson et al., 2004), $k_{OH+HONO} = 6 \times 10^{-12}$
cm$^3$ molecule$^{-1}$ s$^{-1}$ (Atkinson et al., 2004) and $j$(HONO)=$1 \times 10^{-2}$ s$^{-1}$ for an average maximum
noontime OH concentration of $8 \times 10^6$ molecules cm$^{-3}$ (Whalley et al., 2020), NO
concentration of 1.45 ppb (Whalley et al., 2020) and HONO concentration of 0.8 ppb (Whalley
et al., 2020).
The net gas-phase production of HONO from Equation (13 was calculated to be -3.8 ppb hr$^{-1}$
(a net loss) as expected due to HONO loss by photolysis peaking at solar noon, suggesting the
production of HONO heterogeneously from $TiO_2$ and $NO_2$ (~25 ppt hr$^{-1}$) would have little
effect on the overall HONO budget for Beijing summertime at noon.
## 4.2   Production of HONO from photolysis of mixed dust/nitrate aerosols
Oceanic environments, for example the Atlantic Ocean which is impacted by both dust aerosols
from the Sahara and high concentrations of mixed nitrate aerosols from sea spray, and despite
low $NO_2$ concentrations could be important for particulate nitrate photolysis as a source of
HONO (Hanisch and Crowley, 2003;Ye et al., 2017b). From the particulate nitrate photolysis
experiments in the absence of $NO_2$ conducted here, a $j(\text{pNO}_3)_N = (4.73 \pm 1.01) \times 10^{-5}$ s$^{-1}$ was
determined in the presence of the $TiO_2$ photo-catalysts (Section 3.4). Using the experimental
$j$(pNO$_3$), scaled to typical ambient light levels, and a mean noon concentration of nitrate
aerosols of 400 ppt measured at Cape Verde (Reed et al., 2017), taken as an example marine
boundary layer environment with a high concentration of mineral dust aerosols, a rate of
HONO production from particulate nitrate at Cape Verde was calculated as $4.65 \times 10^5$
molecule cm$^{-3}$ s$^{-1}$ (68 ppt hr$^{-1}$). We note that this value would be ~ 50 times smaller for pure
nitrate aerosols. The missing rate of HONO production i.e. not taken into account by the gas
phase production and loss, $P_{other}$, from the Cape Verde RHaMBLe campaign, can be calculated
using the observed HONO concentration, [HONO] and the known gas-phase routes for HONO
production and loss:



$$P_{other} = ([HONO](j(HONO) + k_{OH+HONO}[OH])) - (k_{OH+NO}[OH][NO])) \qquad (14)$$

where $k_{OH+NO} = 3.3 \times 10^{-11}$ cm$^3$ molecule$^{-1}$ s$^{-1}$ (Atkinson et al., 2004), $k_{OH+HONO} = 6 \times 10^{-12}$
cm$^3$ molecule$^{-1}$ s$^{-1}$ (Atkinson et al., 2004) and $j(HONO)=2 \times 10^{-3}$ s$^{-1}$ for average maximum
measured concentrations of $1 \times 10^7$ molecules cm$^{-3}$ for OH (Whalley et al., 2010), $5.41 \times 10^7$
molecule cm$^{-3}$ for NO (Whalley et al., 2010) and $1.23 \times 10^8$ molecule cm$^{-3}$ for HONO (Whalley
et al., 2010).
Using Equation (14) this missing HONO production rate for Cape Verde was 34.6 ppt hr$^{-1}$,
which is within a factor of two of the rate of HONO production (68 ppt hr-1) calculated from
nitrate photolysis using our experimental HONO production data for mixed nitrate/TiO$_2$
aerosols. These results provide further evidence that particulate nitrate photolysis in the
presence of photocatalytic compounds such as TiO$_2$ found in dust could be significant in
closing the HONO budget for this environment (Whalley et al., 2010;Reed et al., 2017;Ye et
al., 2017a).
## 5   Conclusions.
The experimental production of HONO from both illuminated TiO$_2$ aerosols in the presence of
NO$_2$ and from mixed nitrate/TiO$_2$ aerosols in the absence of NO$_2$ was observed, with the
HONO concentrations measured using photo-fragmentation laser-induced fluorescence
spectroscopy. Using experimental data, the reactive uptake of NO$_2$ onto the TiO$_2$ aerosol
surface to produce HONO, $\gamma_{NO_2 \to HONO}$, was determined for NO$_2$ mixing ratios ranging from
34 to 400 ppb, with a maximum $\gamma_{NO_2 \to HONO}$ value of $(1.26 \pm 0.17) \times 10^{-4}$ for single-component
TiO$_2$ aerosols observed at 51 ppb NO$_2$, and for a lamp photon flux of $(1.65 \pm 0.02) \times 10^{16}$
photons cm$^{-2}$ s$^{-1}$ (integrated between 290 and 400 nm). The measured reactive uptake
coefficient, $\gamma_{NO_2 \to HONO}$, showed an increase then subsequent decrease as a function of NO$_2$
mixing ratio, peaking at $51 \pm 5$ ppb. Box modelling studies supported a mechanism involving
two NO$_2$ molecules on the aerosol surface per HONO molecule generated, providing evidence
for the formation of a surface-bound NO$_2$ dimer intermediate. The exact mechanism for HONO
formation, for examples the step(s) which are accelerated in the presence of light, remains
unclear, although previous studies would suggest the process occurs via the isomerisation of
the symmetric N$_2$O$_4$ dimer to give *trans*-ONO-NO$_2$, either via *cis*-ONO-NO$_2$ or directly,
suggested to be more reactive with water than the symmetric dimer (Finlayson-Pitts et al.,
2003;Ramazan et al., 2004;Ramazan et al., 2006;de Jesus Madeiros and Pimentel, 2011;Liu
and Goddard, 2012;Murdachaew et al., 2013;Varner et al., 2014). Investigations into the RH





dependence of the HONO production mechanism on $TiO_2$ aerosols showed a peak in
production between ~25-30 % RH, with lower HONO production at higher $NO_2$ mixing ratios
observed for all RHs tested. The increase in HONO production with increasing RH can be
attributed to a higher concentration of $H_2O$ on the surface increasing its availability for the
hydrolysis reaction to give HONO, whereas a decrease in HONO production after RH ~ 30 %
could be due to the increased water surface concentration inhibiting the adsorption of $NO_2$.
Using the laboratory reactive uptake coefficient for HONO production, $\gamma_{NO_2 \rightarrow HONO}$, the rate of
production of HONO from illuminated aerosols in Beijing in summer for typical $NO_2$ mixing
ratios and aerosol surface areas was found to be similar to that estimated previously for the
production of HONO from urban humic acid aerosol surfaces in Europe.
In the absence of $NO_2$, significant HONO production from 50:50 mixed nitrate/$TiO_2$ aerosols
was measured. Using the experimental HONO concentrations observed, a rate of HONO
production from nitrate photolysis was calculated, which was then scaled to the ambient
conditions encountered at the Cape Verde Atmospheric Observatory in the tropical marine
boundary layer. A HONO production rate of 68 ppt $hr^{-1}$ for the mixed nitrate/$TiO_2$ aerosol was
found for CVAO conditions, similar in magnitude to the missing HONO production rate that
had been calculated previously in order to bring modelled HONO concentrations into line with
field-measured values at CVAO. These results provide further evidence that aerosol particulate
nitrate photolysis may be significant as a source of HONO, and hence $NO_x$, in the remote
marine boundary layer, where mixed aerosols containing nitrate and a photo-catalytic species
such as $TiO_2$, as found in dust, are present.
However, the production of HONO from pure, deliquesced ammonium nitrate aerosols alone
could not be definitively confirmed over the range of conditions used in our experiments,
suggesting that another component within the aerosol is necessary for HONO production.
Future work should be directed towards studying pure nitrate aerosols over a wider range of
conditions, for example varying the aerosol pH, and also adding other chemical species into
the aerosol which may promote HONO production.
*Data availability.* Data presented in this study can be obtained from authors upon request
(d.e.heard@leeds.ac.uk)
*Competing interests.* The authors declare that they have no conflict of interest.





*Acknowledgements*. We are grateful to the Natural Environmental Research Council for
funding a SPHERES PhD studentship (Joanna E. Dyson) and for funding the EXHALE project
(grant number NE/S006680/1).

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
