# Peer review of "aerosol particles and following the illumination of mixed TiO2/ammonium nitrate particles."

_Atmospheric Chemistry and Physics, 2020_

## Short Comment (SC1) · 4 Dec 2020

This short comment is intended simply to inform the authors of the existence of fairly extensive previous studies of the interaction of NO2 with TiO2 under UV irradiation:

El Zein, A.; Bedjanian, Y., Interaction of NO2 with TiO2 surface under UV irradiation: measurements of the uptake coefficient. Atmos. Chem. Phys. 2012, 12 (2), 1013-1020.

Bedjanian, Y.; El Zein, A., Interaction of NO2 with TiO2 Surface Under UV Irradiation:

[Figure]

Products Study. J. Phys. Chem. A 2012, 116 (7), 1758-1764.

These studies report the measurements of the uptake coefficient and distribution of the reaction products (not only HONO, but also NO and N2O) as a function of irradiance intensity, relative humidity, temperature and concentrations of NO2 and O2, i.e. they are directly related to the subject of the authors' paper and could at least be mentioned in the manuscript. By the way, the possible secondary reaction of HONO with TiO2 which can occur in the reactive system was also investigated in our group:

El Zein, A.; Bedjanian, Y., Reactive uptake of HONO to TiO2 surface: "dark" reaction. J. Phys. Chem. A 2012, 116 (14), 3665-3672.

El Zein, A.; Bedjanian, Y.; Romanias, M. N., Kinetics and products of HONO interaction with TiO2 surface under UV irradiation. Atmos. Environ. 2013, 67 (0), 203-210.

---

## Author Comment (AC1) · 4 Dec 2020

We thank Dr Bedjanian for his comment, and for informing us of those studies. Our paper is focussed on HONO production at TiO2 aerosol surfaces and so our citations were directed more towards those types of studies, rather than for TiO2 surfaces more generally. However, we should have cited these studies of Dr Bedjanian, as they are related to the work of our paper, and so we will incorporate those references in the next version of the paper, and include an appropriate discussion. Thank you again.

---

## Referee Comment (RC1) · Anonymous Referee #1 · 3 Jan 2021

In this study the authors used an aerosol flow tube reactor connected to a photo-fragmentation laser induced fluorescence detection set-up to evaluate the heterogeneous chemistry of NO2 with irradiated TiO2 aerosols. The uptake coefficients of NO2 were determined for NO2 mixing ratios ranging between 34 and 400 ppb. The HONO production was determined as well at different relative humidities (RH), the highest being at 25 % RH. The performed kinetic box model suggested HONO production by heterogeneous reaction of NO2 with TiO2 aerosol surface involving two NO2 molecules, and a HONO loss which is dependent on the initial NO2 mixing ratio. Additional experiments have shown that HONO is also formed upon irradiation of mixed nitrate/TiO2 aerosols in the absence of NO2. This is an interesting study following the continuation of a number of previous studies focused on this topic. The experiments are well performed and the kinetic box model was used to support the experimental results. I would suggest publication of this study in Atmospheric Chemistry and Physics as it can be of broad interest for the atmospheric chemistry community.

1) The photo-fragmentation laser induced fluorescence detection apparatus seems promising tool for online measurements of HONO in ambient air. However, the only reference about this instrument is the thesis of Boustead (2019) which is not easily accessible. I wonder if this instrument was previously used in an intercomparison campaign against other well established instruments for real time HONO measurements (e.g. DOAS, LOPAP).

2) The authors observed HONO formation upon irradiation of mixed nitrate/TiO2 aerosols and pure nitrate aerosols but they did not mention in the manuscript whether or not HONO is formed only upon irradiation of TiO2 aerosols in absence of NO2. These tests should be carried out as control experiments.

3) The authors mentioned that the aqueous solutions ready to be dispersed in the air, were obtained by dissolving 5 g of TiO2, but they did not mention the quantity of dissolved ammonium nitrate in the solution. How relevant is this amount of TiO2 dissolved in water?

4) Another very important point is that many papers related to NO2 heterogeneous chemistry on TiO2 as a HONO source are not cited and discussed. For example, Gandolfo et al (Appl. Catal. B: Environ., 2015, 166-167, 84-90; Appl. Catal. B: Environ., 2017, 209, 429-436) have shown that the disproportionation reaction of NO2, which has been also suggested as a night-time source of HONO in the atmosphere, can be photocatalytically enhanced in the presence of TiO2 which is in agreement with the statement in this study that two NO2 molecules forming HONO are required

to reproduce the experimental trend of the uptake coefficients and observed HONO concentrations. Furthermore, a similar profile of the observed dependence of HONO mixing ratios with the RH was also observed by Gandolfo et al. (2015) by detecting a maximum of HONO at 30 % RH as was measured in this study. Increase of HONO with RH on building surface containing TiO2 was also observed by Langridge et al (Atmospheric Environment 43 (2009) 5128-5131).

―――――――――――――――――――――――

---

## Referee Comment (RC2) · Anonymous Referee #2 · 15 Jan 2021

The article by Dyson et al. describes a laboratory study of the efficiency of the chemical transformation of NO2 into HONO by aerosol particles, and the release of HONO from an aerosol containing TiO2 and ammonium nitrate, compounds commonly found within tropospheric aerosol. The title reflects only the first part of the study so probably should be amended.

HONO formation from NO2 is an important process for atmospheric chemistry, with implications for the free-radical budget of the troposphere. The second area, the release of HONO from nitrate-containing mineral dust aerosols, may be important if there is

TiO2 present in the mineral dust itself. This subject is therefore within the scope of ACP and will be of interest to scientists studying atmospheric free radical budgets, ozone chemistry and atmospheric oxidation lifetimes.

This paper is excellent, being an authoritative quantification of the HONO produced from aerosols (HONO being determined by photolysis of HONO and measurement of resulting OH concentration). It is clearly written and of a very high scientific quality. An aerosol flow-tube is used for the study, with supporting measurements of aerosol size distribution. The manuscript combines an extensive set of flow-tube measurements to determine the efficiency of NO2 to HONO conversion, defined as gamma(NO2-HONO) across a range of relative humidity and NO2 mixing ratios. The measurements are performed at room temperature and pressure. HONO production from TiO2-containing aerosols is quantified as a function of NO2 mixing ratio and relative humidity over the range 12-36%, with HONO production reaching a maximum near 30% RH, and afterwards declining. Observed HONO mixing ratios increase with increasing NO2 mixing ratio up to 50 ppb before declining to a constant value above approx 100 ppb which corresponds to a decreasing HONO $\rightarrow$ NO2 reactive uptake coefficient.

The measurements are discussed in the context of a box model employing three distinct mechanisms and are shown to be reproduced well by the mechanisms, adding further insight. The box model is described well and the manuscript shows the depth of physical chemistry expertise available in this leading group, and provides a valuable review of the chemistry involved which is relevant to the atmosphere.

The study of HONO release from TiO2-containing nitrate aerosols is interesting, but not treated at quite the same depth as the uptake onto TiO2 aerosols. An experiment involving single-component NH4NO3 aerosol was performed at 50% RH, while a second involving (presumably internally mixed) nitrate/TiO2 aerosols was performed at 20% RH. The relative humidity used in this study is on the low side for the boundary layer, particularly the marine boundary layer discussed in this manuscript, and the effect of humidity.

[Figure]

In fact, the only issue I have with the manuscript is the application of the laboratory results to the atmospheric cases mentioned. The authors note the dependence of the HONO production on RH, and even adjust experimental conditions to allow for this. The discussion of the atmospheric implications doesn't discuss the RH effect in much detail, which is a pity, because it may be an important factor, particularly in the May/June case of Beijing, although it would appear not to alter the conclusions of section 4.1, and in the (likely) high relative humidity marine boundary layer in Cape Verde. I would like to see this considered in the revised MS.

Minor points:

Figure 2: according to equation 8, the a plot of k vs SA should pass through the origin, but the plotted data do not appear to. Why is this? Can the authors comment?

L206: what is the time to establish laminar flow? How precisely is the overall interaction time between NO2 and the aerosol surface area known?

L472 and Table1 - the use of a first order rate coefficient to describe the rate of adsorption is interesting, and merits further discussion. In model 1, the use of a constant rate coefficient for this step would imply (for constant sticking probability) a constant surface area. Was the rate coefficient R9 varied between experiments to account for variations in aerosol surface area density?

Figure 10. How was gamma (NO2→HONO) retrieved from the box model?

Section 3.3.2 It would be useful here to identify the key kinetic parameters, that is the ones on which the uptake coefficients most sensitively depend. Given that many of the input kinetic rate coefficients used in the box model have been estimated, it may be useful to show an envelope or other indication of how the uncertainty in the input kinetic rate coefficients propagate through to the calculated uptake coefficients shown in Figure 10.

L637 I'm not clear on why 50% RH was used here when the experiments with the

mixed TiO2/nitrate aerosols were performed at a lower RH.

---

## Author Comment (AC2) · 11 Feb 2021

**Referee 1**

**We thank the referee for their helpful comments. Each comment in turn is shown below followed by our response in bold, and followed by any changes to the manuscript in red.**

In this study the authors used an aerosol flow tube reactor connected to a photofragmentation laser induced fluorescence detection set-up to evaluate the heterogeneous chemistry of NO2 with irradiated TiO2 aerosols. The uptake coefficients of NO2 were determined for NO2 mixing ratios ranging between 34 and 400 ppb. The HONO production was determined as well at different relative humidities (RH), the highest being at 25 % RH. The performed kinetic box model suggested HONO production by heterogeneous reaction of NO2 with TiO2 aerosol surface involving two NO2 molecules, and a HONO loss which is dependent on the initial NO2 mixing ratio. Additional experiments have shown that HONO is also formed upon irradiation of mixed nitrate/TiO2 aerosols in the absence of NO2. This is an interesting study following the continuation of a number of previous studies focused on this topic. The experiments are well performed and the kinetic box model was used to support the experimental results. I would suggest publication of this study in Atmospheric Chemistry and Physics as it can be of broad interest for the atmospheric chemistry community.

1. The photo-fragmentation laser induced fluorescence detection apparatus seems promising tool for online measurements of HONO in ambient air. However, the only reference about this instrument is the thesis of Boustead (2019) which is not easily accessible. I wonder if this instrument was previously used in an intercomparison campaign against other well established instruments for real time HONO measurements (e.g. DOAS, LOPAP).

**We are not aware of any previous intercomparison campaigns that have used this technique. In the revised manuscript we will include a reference to Liao et al., (2006) and also Wang et al. (2020) who have used the photo-fragmentation LIF method for HONO detected during fieldwork. An electronic copy of the PhD thesis of Boustead is available online from the University of Leeds.**

We have revised the manuscript on page 4 using the following text:

"The experimental setup used in this investigation is described in detail in (Boustead, 2019), as well as similar systems having been used to measure HONO in the field (Liao et al., (2006), Wang et al., (2020)), and therefore only a brief description of the setup is given here."

2. The authors observed HONO formation upon irradiation of mixed nitrate/TiO2 aerosols and pure nitrate aerosols but they did not mention in the manuscript whether or not HONO is formed only upon irradiation of TiO2 aerosols in absence of NO2. These tests should be carried out as control experiments.

**There is no significant production of HONO from TiO$_2$ aerosol surfaces without the presence of NO$_2$. We have added the following text:**

Pg 9 ln 229. "Additional experiments showed no significant production of HONO on TiO$_2$ aerosol surfaces without the presence of NO$_2$".

3. The authors mentioned that the aqueous solutions ready to be dispersed in the air, were obtained by dissolving 5 g of TiO2, but they did not mention the quantity of dissolved ammonium nitrate in the solution. How relevant is this amount of TiO2 dissolved in water?

**5g of TiO$_2$ and 5g of ammonium nitrate were dissolved into 500ml milli-Q water, as stated in the manuscript on page 30, line 625. The mass of TiO$_2$ dissolved in water allows some control over the maximum TiO$_2$ which can be atomised into the aerosol phase but does not affect the size distribution of aerosols produced. We get finer control of this by using the HEPA filter.**

4. Another very important point is that many papers related to NO2 heterogeneous chemistry on TiO2 as a HONO source are not cited and discussed. For example, Gandolfo et al (Appl. Catal. B: Environ., 2015, 166-167, 84-90; Appl. Catal. B: Environ., 2017, 209, 429-436) have shown that the

disproportionation reaction of NO2, which has been also suggested as a night-time source of HONO in the atmosphere, can be photocatalytically enhanced in the presence of TiO2 which is in agreement with the statement in this study that two NO2 molecules forming HONO are required to reproduce the experimental trend of the uptake coefficients and observed HONO concentrations. Furthermore, a similar profile of the observed dependence of HONO mixing ratios with the RH was also observed by Gandolfo et al. (2015) by detecting a maximum of HONO at 30 % RH as was measured in this study. Increase of HONO with RH on building surface containing TiO2 was also observed by Langridge et al (Atmospheric Environment 43 (2009) 5128-5131).

**We thank the referee for pointing out those papers. Our paper is focussed on HONO production from the surfaces of suspended TiO$_2$ and other aerosols and so the citations were more aimed towards these types of studies. Previous studies of surface interactions of NO$_2$ to form HONO were only considered when investigating the mechanism of dimer formation. However, we should have cited these studies by Gandolfo, as they are related to the RH dependence studies within our paper. We have included the Gandolfo et al. references in the revised manuscript and also the Langridge et al. paper. These papers are cited as followed in the revised manuscript:**

Page 20, Line 435. "An increase in HONO as a function of RH has also been observed on TiO$_2$-containing surfaces (Gandolfo et al., (2015), Gandolfo et al., (2017), Langridge et al., (2009)) with a similar profile for the observed RH dependence of HONO was observed by Gandolfo et al., (2015) from photocatalytic paint surfaces with a maximum in HONO mixing ratio found at 30 % RH."

**References:**

**Gandolfo, A., Bartolomei, V., Gomez Alvarez, E., Tlili, S., Gligorovski, S., Kleffmann, J., and Wortham, H.: The effectiveness of indoor photocatalytic paints on NOx and HONO levels, App. Catal. B: Environ., 166-167, https://doi.org/10.1016/j.apcatb.2014.11.011, 2015**

**Gandolfo, A., Rouyer, L., Wortham, H., and Gligorovski., D.: The influence of wall temperature on NO$_2$ removal and HONO levels released by indoor photocatalytic paints, App. Catal. B: Environ., 209, https://doi.org/j.apcatb.2017.03.021, 2017**

**Langridge, J. M., Gustafsson, R. J., Griffiths, P. T., Cox, R. A., Lambert, R. M., and Jones, R. L.: Solar driven nitrous acid formation on building material surfaces containing titanium dioxide: A concern for air quality in urban areas?, Atmos. Environ., 43, https://doi.org/10.1016/j.atmosenv.2009.06.046, 2009**

**Liao, W., Hecobian A., Mastromarino, J., and Tan, D.: Development of a photo-fragmentation/laser-induced fluorescence measurement of atmospheric nitrous acid, Atmos. Environ., 40, https://doi.org/10.1016/j.atmosenv.2005.07.001, 2006**

**Wang, C., Bottorff, B., Reidy, E., Rosales, C. M. F., Collins, D. B., Novoselac, A., Farmer, D. K., Vance, M. E., Stevens, P. S., and Abbatt, J. P. D.: Cooking, Bleach Cleaning, and Air Conditioning Strongly Impact Levels of HONO in a House, Environ. Sci. Technol., 54, https://doi.org/10.1021/acs.est.0c05356, 2020**

---

## Author Comment (AC3) · 11 Feb 2021

**Referee 2**

**We thank the referee for their helpful comments. Each comment in turn is shown below followed by our response in bold, and followed by any changes to the manuscript in red.**

The article by Dyson et al. describes a laboratory study of the efficiency of the chemical transformation of NO2 into HONO by aerosol particles, and the release of HONO from an aerosol containing TiO2 and ammonium nitrate, compounds commonly found within tropospheric aerosol. The title reflects only the first part of the study so probably should be amended.

**We agree with the referee and have modified the title of the manuscript to:**

"Production of HONO from NO$_2$ uptake on illuminated TiO$_2$ aerosol particles and following the illumination of mixed TiO$_2$/ammonium nitrate particles".

HONO formation from NO2 is an important process for atmospheric chemistry, with implications for the free-radical budget of the troposphere. The second area, the release of HONO from nitrate-containing mineral dust aerosols, may be important if there is TiO2 present in the mineral dust itself. This subject is therefore within the scope of ACP and will be of interest to scientists studying atmospheric free radical budgets, ozone chemistry and atmospheric oxidation lifetimes. This paper is excellent, being an authoritative quantification of the HONO produced from aerosols (HONO being determined by photolysis of HONO and measurement of resulting OH concentration). It is clearly written and of a very high scientific quality. An aerosol flow-tube is used for the study, with supporting measurements of aerosol size distribution. The manuscript combines an extensive set of flow-tube measurements to determine the efficiency of NO2 to HONO conversion, defined as gamma (NO2-HONO) across a range of relative humidity and NO2 mixing ratios. The measurements are performed at room temperature and pressure. HONO production from TiO2-containing aerosols is quantified as a function of NO2 mixing ratio and relative humidity over the range 12-36%, with HONO production reaching a maximum near 30% RH, and afterwards declining. Observed HONO mixing ratios increase with increasing NO2 mixing ratio up to 50 ppb before declining to a constant value above approx 100 ppb which corresponds to a decreasing HONO → NO2 reactive uptake coefficient. The measurements are discussed in the context of a box model employing three distinct mechanisms and are shown to be reproduced well by the mechanisms, adding further insight. The box model is described well and the manuscript shows the depth of physical chemistry expertise available in this leading group, and provides a valuable review of the chemistry involved which is relevant to the atmosphere. The study of HONO release from TiO2-containing nitrate aerosols is interesting, but not treated at quite the same depth as the uptake onto TiO2 aerosols. An experiment involving single-component NH4NO3 aerosol was performed at 50% RH, while a second involving (presumably internally mixed) nitrate/TiO2 aerosols was performed at 20% RH. The relative humidity used in this study is on the low side for the boundary layer, particularly the marine boundary layer discussed in this manuscript, and the effect of humidity.

In fact, the only issue I have with the manuscript is the application of the laboratory results to the atmospheric cases mentioned. The authors note the dependence of the HONO production on RH, and even adjust experimental conditions to allow for this. The discussion of the atmospheric implications doesn't discuss the RH effect in much detail, which is a pity, because it may be an important factor, particularly in the May/June case of Beijing, although it would appear not to alter the conclusions of section 4.1, and in the (likely) high relative humidity marine boundary layer in Cape Verde. I would like to see this considered in the revised MS.

**As stated by the referee, the average RH in Beijing in summer is higher than used for our RH dependence which as the referee points out showed a decline in HONO production after ~ 30 % RH, although our data are limited in this region. Other studies regarding HONO production on TiO$_2$ aerosols (Gustafsson et al., 2006) and TiO$_2$ containing aerosols (Dupart et al., 2014) also showed a decrease in the uptake coefficient at higher RH. Hence, the NO$_2$ reactive uptake coefficient we use to calculate a production rate for HONO for the conditions in Beijing during summertime are most likely an upper limit. From Gustafsson et al., 2006, we can estimate that**

the uptake coefficient could potentially decrease by as much as 90 % (for pure $TiO_2$) from a relative humidity of ~15 to the 80 % RH which was sometimes experienced in summertime in Beijing. We have changed the text in the manuscript as follows:

Pg 32, line 683. "The average RH in Beijing during summertime is significantly higher than the range of RH used in the $TiO_2$ aerosol experiments. In previous work (Gustafsson., et al, 2007), the $NO_2$ reactive uptake coefficient decreased for relative humidities above those studied here, and hence the HONO production calculated under the conditions in Beijing may represent an upper limit."

Minor Points:

Figure 2: according to equation 8, the a plot of k vs SA should pass through the origin, but the plotted data do not appear to. Why is this? Can the authors comment?

**This is due to a background signal from HONO which is primarily from impurities in the $NO_2$ cylinder used for these experiments. In Figure 4, this background has not been subtracted, and so a comment will be added to the figure caption as follows:**

"**Figure 4.** Pseudo-first-order rate coefficient for HONO production, $k$ (open circles) as a function of aerosol surface area for $[NO_2]$=200 ppb and RH=15 ± 1 %, T = 293 ± 3 K and a photolysis time of 52 ± 2 seconds. The red line is a linear-least squared fit including 1σ confidence bands (dashed lines) weighted to both $x$ and $y$ errors (1σ), the gradient of which yields $\gamma NO_2 \rightarrow HONO$ = (2.17 ± 0.09)× $10^{-5}$ , with the uncertainty representing (1σ). The non-zero y-axis intercept is due to a background HONO signal owing to the presence of a HONO impurity in the $NO_2$ cylinder, and which is not subtracted . The total photon flux of the lamp (see Figure 2 for its spectral output) = (1.63 ± 0.09) × $10^{16}$ photons $cm^{-2}$ $s^{-1}$."

L206: what is the time to establish laminar flow? How precisely is the overall interaction time between NO2 and the aerosol surface area known?

**The distance to establish laminar flow is ~ 29 cm which corresponds to a time of 30 seconds. However, the illuminated section of the flow tube where HONO is generated is the second 50 cm of the tube closest to the HONO detection cell after the laminar flow has been established. The uncertainty in the volumetric flow rate, which is ~3% (controlled by the flow controller output) contributes most to the uncertainty in the illumination time.**

L472 and Table1 - the use of a first order rate coefficient to describe the rate of adsorption is interesting, and merits further discussion. In model 1, the use of a constant rate coefficient for this step would imply (for constant sticking probability) a constant surface area. Was the rate coefficient R9 varied between experiments to account for variations in aerosol surface area density?

**Unlike during the experiments, in which using different surface areas was necessary in order to determine the reactive uptake coefficient, for the modelling studies the aerosol surface area density was kept constant and the impact of changing $NO_2$ was explored. Hence, as only one surface area was used, the rate coefficient for the adsorption of $NO_2$ was kept constant across all $NO_2$ concentrations used in the model, and it was found that as the rate coefficient $k_9$ was lowered this step became the RDS leading to a drop in HONO production.**

Figure 10. How was gamma (NO2→HONO) retrieved from the box model?

**The model outputted the concentrations for HONO at a given illumination time for each initial value of $NO_2$, and γ was then calculated using equations 6 and 7 (Pg 12). However, in the model the aerosol surface area (SA) was kept at a constant value of $1.6 \times 10^{-2}$ $m^2m^{-3}$ as was used in Figure 6, in order to provide a direct comparison with experiment for the dependence of the HONO concentration on initial $NO_2$ concentration. In order to clarify this, we have modified the caption to Figure 10 as follows:**

**Figure 10**. Experimental values (open circles with 1σ error bars), Model 2 (green line) and Model 3 (pink line) calculations for (a) HONO concentration after 52 s illumination and (b) $NO_2$ reactive uptake

coefficient, $\gamma_{NO_2 \rightarrow HONO}$, as a function of the initial $NO_2$ mixing ratio. The mechanisms used for these model runs included a 2:1 stoichiometric relationship between the $NO_2$ adsorbed on the $TiO_2$ aerosol surface and the HONO produced, as well as additional HONO loss reactions which are dependent on $NO_2$, see Table 1 for details. Models 2 and 3 use an Eley-Rideal and Langmuir-Hinshelwood mechanisms, respectively, for the formation of the $NO_2$ dimer on the aerosol surface. Modelled $\gamma_{NO_2 \rightarrow HONO}$ was calculated using Eq. 6 and Eq. 7 with a constant surface area of $1.6 \times 10^{-2}$ $m^2m^{-3}$ chosen to match the aerosol surface area density of $(1.6 \pm 0.8) \times 10^{-2}$ $m^2m^{-3}$ shown in the experimental [HONO] values in (a).

Section 3.3.2. It would be useful here to identify the key kinetic parameters, that is the ones on which the uptake coefficients most sensitively depend. Given that many of the input kinetic rate coefficients used in the box model have been estimated, it may be useful to show an envelope or other indication of how the uncertainty in the input kinetic rate coefficients propagate through to the calculated uptake coefficients shown in Figure 10.

**For models 2 and 3 the shape of the trend in HONO concentration versus $NO_2$ concentration depended strongly on the $NO_2$ dependent loss reaction, R19, whereas the shape of the trend in uptake coefficient, $\gamma$, versus $NO_2$ concentration depended strongly on the choice of a 2:1 stoichiometric ratio of the $NO_2$ molecules adsorbed to the HONO molecules produced. Without these two key processes being included, the outputs of models 2 and 3 look similar to the trends seen in Figure 9 for Model 1 (no maximum in either HONO or $\gamma$ as the $NO_2$ concentration is increased). A third key condition was the requirement that the desorption rate coefficient, $k_{R16}$ = $5 \times 10^{-2}$ $s^{-1}$, had to be larger than the rate coefficient for the loss of HONO, $1 \times 10^{-3}$ $s^{-1}$, but slower than the adsorption rate coefficient, $k_{R9}$, in order to reproduce the trend in HONO versus $NO_2$ seen experimentally.**

**Changing the values of all other kinetic parameters in the model had an effect on the absolute concentration of HONO but crucially not on the shape of the trends in HONO or the uptake coefficient versus $NO_2$ concentration. Changing the values of the rate coefficients for the gas phase loss reactions, R23-27, had a negligible effect on the HONO concentration, whereas changing the rate coefficients for the surface loss processes, R17-18, had more of an effect, whilst still small in comparison to changing the $NO_2$ dependant loss reaction, R19. Loss of $NO_2$ and HONO by gas phase photolysis, R20-21, also only had a small effect on the HONO concentration with the loss of gas phase HONO to the walls, R22, having a very large effect on the absolute concentration, but not on the trend of the HONO concentration or uptake coefficient with $NO_2$ concentration.**

**We performed a sensitivity analysis, during which each rate coefficient for reactions 17-27 was increased by a factor of 5 to see the effect on the [HONO] outputted for Models 2 and 3. The results are shown in the table below:**

| Reaction | Description | Average percentage difference in HONO with an increase in $k$ by a factor of 5 | |
| --- | --- | --- | --- |
| | | Model 2 | Model3 |
| R17 | Surface loss of HONO/ $NO_2$ independent loss reaction | -4 | -5 |
| R18 | Surface loss of HONO/ $NO_2$ independent loss reaction | -2 | -4 |
| R19 | $NO_2$ dependant loss reaction | -154 (% diff increased with increasing $NO_2$ ranging from -0.006 to -451) | -114 (% diff increased with increasing $NO_2$ ranging from -0.003 to -447) |

| | | | |
|---|---|---|---|
| R20 | NO$_2$ gas phase photolysis | 11 (% diff increased with increasing NO$_2$ ranging from -2.5 to 44) | 12 (% diff increased with increasing NO$_2$ ranging from -0.93 to 53) |
| R21 | HONO gas phase photolysis | -17 | -10 |
| R22 | Gas phase HONO wall loss | -314 | -270 |
| R23 | Gas phase loss of HONO | 0 | 0 |
| R24 | Gas phase loss of NO$_2$ | 0 | 0 |
| R25 | Reactions of triplet oxygen / Gas phase loss of NO$_2$ | 0 | 0 |
| R26 | Reactions of triplet oxygen | 0 | 0 |
| R27 | Reactions of triplet oxygen | 0 | 0 |

In order to aid understanding of the key kinetics parameters, we have replaced lines 558-563 with the following text in the manuscript:

Page 27, line 558. "For models 2 and 3 the shape of the trend in HONO concentration and uptake coefficient, $\gamma$, versus NO$_2$ concentration depended strongly on the value of $k_{R19}$ reaction, R19, and the choice of a 2:1 stoichiometric ratio of the NO$_2$ molecules adsorbed to the HONO molecules produced. Without these two key processes being included, a maximum in either the HONO concentration or $\gamma$ as the NO$_2$ concentration is increased could not be obtained in the model. A third key condition was the requirement that the desorption rate coefficient, $k_{R16}$, be larger than the rate coefficient for the loss of HONO, $k_{R17}$ and $k_{R18}=1 \times 10^{-3}$ s$^{-1}$, but slower than the adsorption rate coefficient, $k_{R9}$. Changing the values of all other kinetic parameters in the model had an effect on the absolute concentration of HONO, but crucially not on the shape of the trends in HONO or the uptake coefficient versus NO$_2$ concentration. Changing the values of the rate coefficients for the gas phase loss reactions, R23-27, only had a very small impact on the HONO concentration."

L637 I'm not clear on why 50% RH was used here when the experiments with the mixed TiO2/nitrate aerosols were performed at lower RH.

**Sorry, this is a typographical error. The calculations were done at the same RH as the experiment was performed at. The text of the revised manuscript has been made modified as follows:**

Pg 30 ln 637. "Using the Aerosol Inorganic Model (AIM) (Clegg et al., 1998; Wexler and Clegg, 2002), the nitrate content of the aerosol at both 20 and 50% RH was calculated, in accordance with the experimental RH conditions."